# TEST-TIME ADAPTATION FOR IMAGE COMPRESSION WITH DISTRIBUTION REGULARIZATION

**Kecheng Chen, Pingping Zhang, Tiexin Qin, Shiqi Wang, Hong Yan, Haoliang Li**[*]
City University of Hong Kong, Hong Kong SAR. [*]Corresponding author.
`cs.ckc96@gmail.com; haoliang.li@my.cityu.edu.hk`

## ABSTRACT

Current test- or compression-time adaptation image compression (TTA-IC) approaches, which leverage both latent and decoder refinements as a two-step adaptation scheme, have potentially enhanced the rate-distortion (R-D) performance of learned image compression models on cross-domain compression tasks, *e.g.,* from natural to screen content images. However, compared with the emergence of various decoder refinement variants, the latent refinement, as an inseparable ingredient, is barely tailored to cross-domain scenarios. To this end, we aim to develop an advanced latent refinement method by extending the effective hybrid latent refinement (HLR) method, which is designed for *in-domain* inference improvement but shows noticeable degradation of the rate cost in *cross-domain* tasks. Specifically, we first provide theoretical analyses, in a cue of marginalization approximation from in- to cross-domain scenarios, to uncover that the vanilla HLR suffers from an underlying mismatch between refined Gaussian conditional and hyperprior distributions, leading to deteriorated joint probability approximation of marginal distribution with increased rate consumption. To remedy this issue, we introduce a simple Bayesian approximation-endowed *distribution regularization* to encourage learning a better joint probability approximation in a plug-and-play manner. Extensive experiments on six in- and cross-domain datasets demonstrate that our proposed method not only improves the R-D performance compared with other latent refinement counterparts, but also can be flexibly integrated into existing TTA-IC methods with incremental benefits. Our code is available at https://tonyckc.github.io/TTA-IC-DR/.

## 1 INTRODUCTION

With rapid developments in data streaming techniques, fruitful high-resolution images need to be transmitted online between edge devices. It is therefore imperative to develop more efficient, effective, and versatile image compression approaches for better storage and transmission. To this end, we have witnessed learning-based image compression (LIC) methods (Ballé et al., 2016; 2018; Cheng et al., 2020; Kim et al., 2024) significantly outperform conventional codecs, such as VVC (Bross et al., 2021) and JPEG (Wallace, 1991) for rate-distortion (R-D) performance. Such gains mainly derive from the unprecedented non-linear transform capacity of deep neural networks (DNN) and accurate probability representations for entropy coding in an end-to-end R-D cost-guided learning framework.

Nevertheless, these DNN-based LIC approaches inevitably inherit the identically and independently distributed (*i.i.d.*) assumption between the source (training data) and target (testing data) domains, which may not always hold in versatile image compression scenarios, *e.g.,* there is a significant distribution gap between natural and screen content images. Compared with in-domain compression (*i.e., i.i.d.* assumption holds), such *domain shifts* would deteriorate the R-D performance of DNN-based codecs in cross-domain compression. For example, most advanced LIC models leverage a hyperprior-based entropy framework (Ballé et al., 2018), where the hyperprior model extracts the side information $z$ of the latent variable $y$ to capture Gaussian conditional probability $p(y|z)$ for coding, and the side information is coding by the hyperprior probability $p(z)$ learned from entropy bottleneck. In cross-domain scenarios, the discrepancy in statistical property between source and target domains will cause inaccuracy or ineffectiveness of learned probability models (Ulhaq & Bajić, 2024), leading to suboptimal entropy coding with additional bit consumption. Moreover, it is difficult for a source-trained decoder to render high-fidelity target images due to domain shifts.

To this end, it is necessary to conduct a test- or compression-time adaptation for image compression (TTA-IC). One approach (Zou et al., 2021) is to update the decoder of the entropy model, which involves the transmission of adapted parameters to the decoder side. Although recent parameter-efficient transfer learning-based (Hu et al., 2021) extensions (Shen et al., 2023) reduce the huge bitrate overhead of parameter transmission to an acceptable level, the entropy optimization of updated parameters and extra bit consumption are still troublesome. Another promising approach (Djelouah & Schroers, 2019) is to directly refine the latent variables without altering any model parameters. Compared with the first approach, such a branch is still effective for many specialized neural decoders, where model parameters are hard-coded and non-modified (Dass et al., 2023).

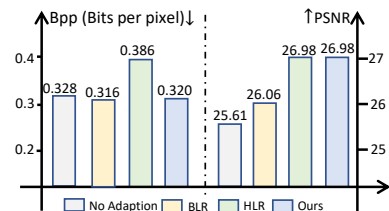

Figure 1: Comparison of various latent refinement methods under cross-domain tasks (SIQAD screen content dataset). The Cheng20 (Cheng et al., 2020) (quality= 0, pre-trained on natural images) model is used.

Here, we focus on the second approach, *i.e.,* latent refinement, due to its simple optimization, friendly bitrate, and natural immunity to catastrophic forgetting. Some state-of-the-art (SOTA) TTA-IC methods (Campos et al., 2019; Shen et al., 2023) actually have introduced basic latent refinement (BLR) (Djelouah & Schroers, 2019) into their two-step adaptation framework including latent and decoder refinements, *e.g.,* updating the latent variable $y$ by R-D objective. Yet, the Gaussian conditional probability $p(y|z)$ and hyperprior $p(z)$ are still inaccurate without refining the side information $z$, leading to suboptimal cross-domain R-D performance. A potential solution may be hybrid latent refinement (HLR) (Yang et al., 2020) that conducts a joint update of latent variable $y$ and side information $z$. Although the vanilla HLR, tailored to *in-domain* inference improvement, exhibits better R-D performance compared with BLR in *in-domain* TTA-IC tasks (Yang et al., 2020), it suffers from a significant downside in rate cost while enhancing the reconstruction quality in *cross-domain* scenarios, as depicted in Figure 1. This degradation in rate cost is, however, ignored by existing two-step TTA-IC approaches (Lv et al., 2023; Tsubota et al., 2023; Shen et al., 2023) that directly impose the vanilla HLR as their inseparable ingredient.

Motivated by the abovementioned analyses, this study aims to develop an advanced latent refinement method that can be adaptive to cross-domain TTA-IC with *consistent* R-D gains. Such a scheme can not only render better latent representations with *zero* model update and transmission but also enhance the R-D performance of existing SOTA TTA-IC approaches as an effective alternative to existing latent refinement. With these goals in mind, we propose to tailor the vanilla HLR method designed for *in-domain* inference improvement to *cross-domain* cases. To achieve this, a theoretical analysis, in a cue of marginalization approximation from in- to cross-domain scenarios, is provided to reveal the degradation reasons of the vanilla HLR in the cross-domain scenario. In a nutshell, we uncover that the underlying mismatch between refined Gaussian conditional and hyperprior distributions may trigger the deteriorated joint probability approximation of marginal distribution, leading to increased rate consumption. To remedy this issue, we introduce a novel *distribution regularization* to the existing R-D objective, which encourages learning a better joint probability approximation from a theoretical perspective. Moreover, we impose a Bayesian approximation of the proposed distribution regularization to circumvent any model modification in a plug-and-play manner. Experiments on in- and cross-domain tasks demonstrate that our proposed method surpasses other latent refinements approaches and contributes to SOTA TTA-IC models. Our contributions are summarized as follows.

- To the best of our knowledge, we are the first to reveal the degradation reasons of advanced LIC and corresponding latent refinement methods in a marginalization approximation perspective.
- We propose a Bayesian approximation-endowed distribution regularization to encourage learning a better joint probability approximation in a plug-and-play manner.
- The experimental results demonstrate our proposed method not only surpasses existing latent refinement counterparts but also can be integrated into SOTA TTA-IC methods.

## 2 RELATED WORKS

Various TTA approaches have been proposed to tackle distribution shifts at test time. For instance, Wang et al. (2020) introduced TENT that updates normalization statistics and optimizes channel-wise affine transformations. Niu et al. (2022) adapted models based on active sample selection and a Fisher

regularize. Darestani et al. (2022) combined self-supervision during training with test-time training. Recent works extend TTA to handle more challenging scenarios in image restoration tasks, such as image supervision (Deng et al., 2023) and blind image quality assessment (Roy et al., 2023). Notably, Xu et al. (2023) is the first attempt to optimize the downscaled representation instead of the model.

To address the domain gap in TTA-IC tasks, various studies have explored strategies like fine-tuning the encoder during inference to refine latent variables without extra bit transmission. For example, Djelouah & Schroers (2019) introduced BLR, which uses gradient descent on latent representations guided by R-D cost, while Guo et al. (2020) extended BLR with a two-step approach optimizing both latent variables and side information. Beyond BLR, Yang et al. (2020) proposed HLR, which jointly refines latent variables and side information while eliminating discretization gaps. While HLR improves R-D performance for *in-domain* tasks, its effectiveness in *cross-domain* scenarios remains unclear. Decoder refinement is another popular approach, where decoder parameters are updated and transmitted during test-time (Lam et al., 2020; Zou et al., 2021), though it often incurs higher bit costs (Kim et al., 2024). Recent SOTA TTA-IC methods inspired by parameter-efficient transfer learning (PETL) introduced decoder adaptors to improve reconstruction with lower bit consumption (Shen et al., 2023; Lv et al., 2023; Tsubota et al., 2023), as well as combining either BLR or HLR as the first step of their two-step framework. In a nutshell, compared with the emergence of various PETL-based decoder refinement variants, the latent refinement is barely tailored to cross-domain scenarios, thereby exhibiting underexplored space.

## 3 METHODOLOGY

We first explain existing learned image compression from a perspective of marginalization approximation in the context of in-domain R-D cost-guided training. Then, we analyze the practical marginalization approximation when applying the learned image codec to cross-domain image compression. We further render a simple solution to achieve better R-D performance for cross-domain image compression based on the analysis of practical marginalization approximation.

**Preliminary**. Existing advanced LIC approaches mainly adopt a hyperprior-based entropy framework (Ballé et al., 2018). As illustrated in Figure 2(a), given an image $x$, an analysis transform $g_a$ can compute a latent representation $y=g_a(x)$, which is then quantized and transmitted to the receiver in the context of an entropy model $p(y)$. A synthesis transform $g_e$ can finally render the reconstruction. To enhance the bitrate, a hyperprior is typically introduced to approximate the entropy model $p(y)$ (marginal distribution) as a joint probability, which involves a hyper latent variable $z = h_a(y)$ (*a.k.a.* side information) calculated by a hyper analysis transform $h_a$, and a hyper synthesis $h_e$ usually output the mean and scale of a Gaussian distribution as the entropy model $p(y|z)$ for the entropy coding of latent representation $y$. One line of LIC works aims to achieve better joint probability approximation of marginal distribution by various model elaborations (*e.g.,* self-attention (Zou et al., 2022) and autoregressive (Minnen & Singh, 2020)). Our work mainly analyzes the marginalization approximation from in- to cross-domain and proposes a solution for better cross-domain performance.

### 3.1 IN-DOMAIN MARGINALIZATION APPROXIMATION

First, we introduce the marginalization approximation used by existing entropy models (Ballé et al., 2018; Cheng et al., 2020; Zou et al., 2022). Specifically, in the context of hyperprior-based entropy models, the following result holds,

**Lemma 1** (Ballé et al. (2018)). *Let $p(y|z)$ and $p(z)$ be accessed, a joint probability over $y$ and $z$ can be constructed to approximate the true marginal probability over $y$,*

$$p(y) = \int p(y,z)dz, \quad p(y,z) = p(y|z) \cdot p(z), \quad s.t., y = g_a(x), z = h_a(y), \quad (1)$$

Lemma 1 implies that the compression of the raw image involves the compression of $z$ using the learned prior distribution $p(z)$ and further compression of $y$ using the learned conditional distribution $p(y|z)$. Practically, we implement $p(z)$ by a non-parametric, fully-factorized density model (*a.k.a.*, entropy bottleneck) (Ballé et al., 2016), and implement $p(y|z)$ by a mean-scale Gaussian model, *i.e.,*

$$p(z) = p(z|\varphi), p(y|z) = p(y|z, \theta_{h_e}) = \mathcal{N}(\mu, \sigma), \mu, \sigma = h_e(z; \theta_{h_e}), \quad (2)$$

where $\varphi$ denotes the learnable parameters of the density model characterized by the univariate distribution, and $h_e(\cdot)$ denotes a synthesis transform function of the hyperprior model parameterized

by learnable $\theta_{h_e}$. The learning of these parameterized probability models derives from training data, which means that the *i.i.d.* assumption between training and testing data is implicitly needed. We therefore call such marginalization approximation as the *in-domain* one.

The widely used R-D objective can also be formulated by maximizing the probability of joint distribution $p(y, z)$ and the posterior distribution $p(x|y, \theta_{g_e})$, *i.e.*, $\mathcal{L}_{rd} =$

$$\mathbb{E}_{x \sim p(x), y, z \sim p(y,z)}[-\log[p(y|z) \cdot p(z)] + \lambda(-\log p(x|y))] = \mathbb{E}_p[\underbrace{-\log p(y|z) - \log p(z)}_{Rate} + \underbrace{\lambda\|x - g_e(y)\|_2^2}_{Distortion}]. \quad (3)$$

Minimizing Eq. (3) can contribute to an optimal joint probability $p(y, z)$, which approximates the marginal distribution $p(y)$, *i.e.*, Lemma 1 holds. Formally, the optimal probability representation is defined as follows.

**Definition 1.** *(Optimal Probability Representation). By minimizing Eq. 3, there exist learnable parameters $\theta_{h_e}^*$ and $\varphi^*$ that achieve minimum rate cost for $p(y|z)$ and $p(z)$. According to Shannon's entropy theorem, the learned encoding distributions $p(y|z)$ and $p(z)$ thus converge to optimal probability representations $p(y^*|z^*)$ and $p(z^*)$ that exactly match the true underlying distributions:*

$$p(y \mid z) = p(y \mid z, \theta_{h_e}^*) = p(y^* \mid z^*) \quad p(z) = p(z \mid \varphi^*) = p(z^*).$$

However, such marginalization approximation requires more information to be encoded (Townsend et al., 2019). Theoretically, we can quantify extra encoding information of marginalization approximation using the entropy $H(\cdot)$ of distribution as the rate consumption, *i.e.*,

**Proposition 1.** *Let $y$ and $z$ be the latent and hyper latent variables, and these variables with the asterisk be their optimal representations. In the context of in-domain image compression, if an optimal joint probability approximation of true marginal distribution can be achieved by minimizing Eq. (3), the extra rate consumption of marginalization approximation is*

$$\Delta H^* = H(y, z) - H(y^*) = \mathbb{E}_{y,z \sim p(y,z)}[-\log p(z|y)], \quad (4)$$

*Proof.* On the one hand, with joint probability and the Bayesian rule $p(y^*) = \frac{p(y^*|z^*)p(z^*)}{p(z^*|y^*)}$, we have

$$H(y, z) = \mathbb{E}_p[-\log p(y|z) - \log p(z)], \quad H(y^*) = \mathbb{E}_p[-\log p(y^*|z^*) - \log p(z^*) + \log p(z^*|y^*)]. \quad (5)$$

Then, we have

$$\Delta H^* = \mathbb{E}_p[-\log p(y|z) - (-\log p(y^*|z^*))] + \mathbb{E}_p[-\log p(z) - (-\log p(z^*))] + \mathbb{E}_p[-\log p(z^*|y^*)]. \quad (6)$$

On the other hand, as there exist optimal probability representations for $p(y|z)$ and $p(z)$ for in-domain image compression by minimizing Eq. (3), we have

$$\mathbb{E}_p[-\log p(y|z) - (-\log p(y^*|z^*))] = 0, \quad s.t. \quad p(y|z) = p(y^*|z^*) \text{ and } p(z) = p(z^*), \quad (7)$$

$$\mathbb{E}_p[-\log p(z) - (-\log p(z^*))] = 0, \quad s.t. \quad p(z) = p(z^*) \quad (8)$$

Thus,

$$\Delta H^* = \mathbb{E}_{y^*, z^* \sim p(y^*, z^*)}[-\log p(y^*|z^*)] = \mathbb{E}_{y,z \sim p(y,z)}[-\log p(z|y)] \quad (9)$$

For in-domain image compression, Eq. (7), Eq. (8), and Eq. (9) hold, as $p(y|z)$, $p(z)$, and $p(z|y)$ are close to optimal probability representations $p(y^*|z^*)$, $p(z^*)$, and $p(z^*|y^*)$ due to the assumption of the optimal joint probability approximation of true marginal distribution. □

Lemma 1 and Prop. 1 are mainly adaptive to in-domain image compression. By this cue, we will discuss practical marginalization approximation and its impact on extra rate consumption in the context of cross-domain image compression.

### 3.2 GENERALIZE TO CROSS-DOMAIN MARGINALIZATION APPROXIMATION

When we apply the learned image codec to cross-domain scenarios, *e.g.*, assuming the source-domain image as $x_s$ and the target-domain image $x_t$, the *i.i.d.* assumption between source and target domains violates. We can derive that Lemma 1 may not hold due to the following insight:

**Proposition 2.** *The practical joint probability $p(y_t, z_t)$ on cross-domain images will deteriorate to make it not equivalent to an optimal joint probability $p(y_t^*, z_t^*)$ (that exactly match the true underlying distributions, in other words, $p(y_t^*|z_t^*)$ and $p(z_t^*)$ are also optimal as in Definition 1) that also corresponds to an optimal marginal approximation distribution $p(y_t^*)$, leading to deteriorated marginalization approximation, i.e.,*

$$p(y_t^*, z_t^*) \neq p(y_t, z_t), p(y_t^*) \not\approx p(y_t, z_t), p(y_t, z_t) = p(y_t|z_t, \theta_{h_e}^s) \cdot p(z_t|\varphi^s), \quad (10)$$

*Proof.* It is straightforward to induce the proof of this proposition based on commonly used conclusions and existing observations. Specifically, the distribution shifts, *i.e.,* $p(x_t) \neq p(x_s)$, would result in the entropy bottleneck $p(z_t|\varphi^s)$ parameterized by $\varphi^s$ (which replaces $\varphi$ in Eq. (2) to emphasize its correlation with source-domain images $x_s$) is quite poor at specializing and encoding $z_t$ for cross-domain images (Ulhaq & Bajić, 2024). Also, $p(y_t|z_t, \theta^s_{h_e})$ is inaccurate to encode $y_t$ due to source image-correlated $\theta^s_{h_e}$ and unreliable $z_t$(Campos et al., 2019). As the deterioration of independent distributions results in the deterioration of the joint distribution, the joint probability $p(y_t, z_t)$ significantly deteriorates compared with the optimal one $p(y_t^*, z_t^*)$ and further causes an unfavorable marginalization approximation $p(y_t^*) \not\approx p(y_t, z_t)$. □

In light of Prop. 2, a cross-domain extension of Prop. 1 can be derived as follows,

**Corollary 1.** *The extra rate consumption of cross-domain marginalization approximation $\Delta H$ will be larger than that of in-domain marginalization approximation, i.e.,, $\Delta H > \Delta H^*$, due to deteriorated joint probability.*

*Proof.* Due to distribution shifts and suboptimal source-domain parameters $(\theta^s_{h_e}, \varphi^s)$, Eqs. (7) and (8) in Prop. 1 can be further represented based on Prop. 2 as follows,

$$\mathbb{E}_p[-\log p(y_t|z_t, \theta^s_{h_e}) - (-\log p(y_t^*|z_t^*))] > 0, \quad s.t. \quad p(y_t|z_t, \theta^s_{h_e}) \neq p(y_t^*|z_t^*) \text{ and } p(z_t|\varphi^s) \neq p(z_t^*),$$
(11)

$$\mathbb{E}_p[-\log p(z_t|\varphi^s) - (-\log p(z_t^*))] > 0, \quad s.t. \quad p(z_t|\varphi^s) \neq p(z_t^*),$$
(12)

where $p(y_t|z_t, \theta^s_{h_e}) \neq p(y_t^*|z_t^*)$ holds due to source image-correlated $\theta^s_{h_e}$ and unreliable $z_t$. $p(z_t|\varphi^s) \neq p(z_t^*)$ holds due to poor specialization of entropy bottleneck $\varphi^s$. Let $p(y_t|z_t, \theta^s_{h_e}) = p(y_t|z_t)$ and $p(z_t|\varphi^s) = p(z_t)$ for brevity. We have $\Delta H =$

$$\mathbb{E}_p[-\log p(y_t|z_t) - (-\log p(y_t^*|z_t^*))] + \mathbb{E}_p[-\log p(z_t) - (-\log p(z_t^*))] + \mathbb{E}_p[-\log p(z_t^*|y_t^*)] > \Delta H^* \quad (13)$$

Since optimal probabilities always match the true underlying distributions from Prop. 2, $p(z_t^*|y_t^*)$ is equivalent to $p(z^*|y^*)$ in Eq. 9 for optimal posterior distributions. Thus, Eq.13 holds and implies more rate consumption is potentially incurred in cross-domain scenarios. □

Besides, distortion error is increased as the posterior probability $p(x_t|y_t, \theta^s_{g_e})$ correlated with source-domain images $x_s$ is suboptimal for cross-domain images $x_t$. To enhance the R-D performance on cross-domain images $x_t$, latent refinement is proposed to optimize latent variables while making model parameters unchanged. We summarize existing latent refinement methods as below.

Assume that the initial latent representations $y_t$ obtained by analysis transform as $y_t^0$ and the initial latent representations $z_t$ obtained by hyperprior transform as $z_t^0$. The BLR scheme (refers to Figure 2) only updates the latent representation $y_t^0$ with M > 1 steps, for each step $m$:

$$y_t^{m+1} = y_t^m - \epsilon \cdot \frac{\partial \mathcal{L}_{blr}}{\partial y_t}, \mathcal{L}_{blr} = \mathbb{E}_{x_t \sim p_{x_t}, y_t, z_t \sim p_{y_t, z_t}}[-\log p(y_t^m|z_t^0, \theta^s_{h_e}) + \lambda(-\log p(x_t|y_t^m, \theta_{g_e}))], \quad (14)$$

where the R-D cost excludes $-\log p(z)$ in Eq. (3) as the latent representation $z_t^0$ is unchanged. For BLR, the gain of R-D performance is limited, as it is difficult to obtain an updated $y_t^m$ that can simultaneously lead to minimal rate cost and minimal distortion error in Eq. (14) due to fixed $z_t^0$. Instead, the HLR in (Yang et al., 2020) not only conducts a joint amortization gap minimization but also eliminates the discretization gap for latent representations using the following optimization step,

$$y_t^{m+1} = y_t^m - \epsilon \cdot \frac{\partial \mathcal{L}_{hlr}}{\partial y_t}, z_t^{m+1} = z_t^m - \epsilon \cdot \frac{\partial \mathcal{L}_{hlr}}{\partial z_t},$$

$$\mathcal{L}_{hlr} = \mathbb{E}_{x_t \sim p_{x_t}, y_t, z_t \sim p_{y_t, z_t}}[-\log p(y_t^m|z_t^m, \theta^s_{h_e}) - \log p(z_t^m|\varphi^s) + \lambda(-\log p(x_t|y_t^m))]. \quad (15)$$

By simultaneously optimizing $y_t$ and $z_t$, Yang et al. (2020) reported significant R-D gains compared with BLR for in-domain adaptive compression. However, when we impose the HLR scheme to cross-domain adaptive compression, such gains are marginal as shown in Figure 1, where the reconstruction quality of the HLR achieves significant gains compared with BLR, but the HLR consumes more bits.

***Discussion*** *– we provide the analyses of degradation reasons of the vanilla HLR in the cross-domain scenario using Cor. 1 and Prop. 2. First, compared with in-domain compression, the biggest trouble in the cross-domain scenario is that the static entropy bottleneck correlated with the distribution property of source-domain images cannot well model the true distribution of cross-domain latent variable $z_t$ as in Prop. 2. Second, although updating $z_t^0$ to an appropriate $z_t^m$ that can render a minimal negative log-likelihood of $p(z_t^m|\varphi^s)$ is possible, ensuring $z_t^m$ as an appropriate Gaussian condition*

of $y_t^m$ is questionable for a minimal negative log-likelihood of $p(y_t^m|z_t^m, \theta_{h_e}^s)$. Finally, by recalling Porp. 2 and Cor. 1, more bit consumption by the HLR implies that the finally updated joint probability $p(y_t^M, z_t^M)$ may be further away from the optimal (unknown) joint probability $p(y_t^*, z_t^*)$, and even the initial joint probability $p(y_t^0, z_t^0)$. This can be directly reflected by more rate consumption of marginal approximation by extending the result of Cor. 1 to specific update steps,

$$\Delta H^M > \Delta H^0 > \Delta H^*. \tag{16}$$

To conclude, the underlying mismatch between $p(z_t^m|\varphi^s)$ and $p(y_t^m|z_t^m, \theta_h^s)$ (in the first two points) triggers deteriorated joint probability approximation of true marginal distribution (in the third point).

## 3.3 DISTRIBUTION REGULARIZATION VIA BAYESIAN APPROXIMATION

Thus, we are interested in an advanced latent refinement method, which not only can significantly improve the reconstruction quality but also can enjoy a mild rate cost after latent refinement. Motivated by cross-domain joint probability approximation in Prop. 2 and corresponding impact for rate consumption in Cor. 1, we propose to introduce a simple yet efficient distribution regularization into the objective of vanilla HLR as follows, $\mathcal{L}_{DR} =$

$$\mathbb{E}_{x_t \sim p_{x_t}, y_t, z_t \sim p_{y_t, z_t}}[-\log p(y_t^m|z_t^m, \theta_{h_e}^s) - \log p(z_t^m|\varphi^s) + \underbrace{\beta(-\log p(z_t^m|y_t^m))} + \lambda(-\log p(x_t|y_t^m))], \tag{17}$$

where the third term is the distribution regularization, which serves as an empirical approximation of optimal posterior distribution $p(z_t^*|y_t^*)$ that can be optimized directly using this objective. The proposed distribution regularization has two advantages as follows.

(i) — If the balance coefficient $\beta$ is 1, minimizing the entropy of probability estimates (*i.e.*, the first two terms in Eq. (17)) is equivalent to minimizing the distribution gap between estimated probability $p(y|z)$ or $p(z)$ and (unknown) optimal probability $p(y^*|z^*)$ or $p(z^*)$ (MacKay, 2003). This coincides with the objectives of the first two terms in Eq. (13) of Cor. 1. Thus, $\mathcal{L}_{DR}$ can be derived as follows,

$$\mathcal{L}_{DR} \propto \Delta H + \lambda(-\log p(x_t|y_t^m)). \tag{18}$$

By recalling Eq. (13) in Cor. 1 and Eq. (16), Eq. (18) implies that the minimization of extra rate consumption of marginalization approximation corresponds to encouraging the deteriorated joint probability approximation to approach the initial and even optimal ones. In other words, $\Delta H^*$ is the *lower bound* of the first term of Eq. (18), *i.e.*, the better joint probability approximation, the closer to the lower bound. $\mathcal{L}_{DR}$ can potentially remedy the additional rate of consumption of vanilla HLR.

(ii) — For the optimization process in Eq. (15), there is no explicit constraint to ensure that $z_t^m$ can well match its posterior distribution un-

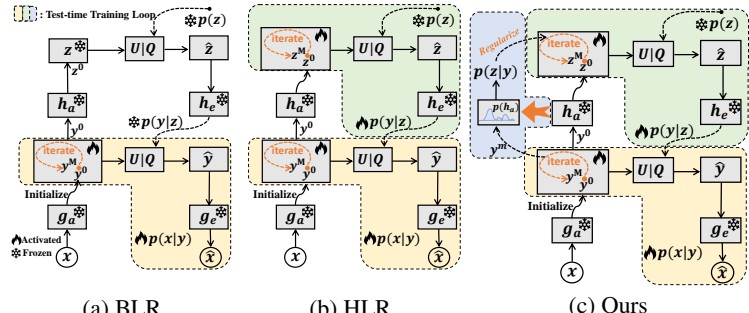

(a) BLR      (b) HLR      (c) Ours

Figure 2: Architectures of different latent refinement TTA-IC methods. $U|Q$ represents the quantization and entropy coding.

der the condition of $y_t^m$, leading to deteriorated joint probability. As a practical implementation of the ideal posterior distribution, the proposed distribution regularization can eliminate such an issue.

To implement the $\mathcal{L}_{DR}$, it is necessary to model the estimated posterior distribution $p(z_t^m|y_t^m)$ of $z_t^m$ given $y_t^m$. However, for existing hyperprior models, the practical $\hat{z}_t^m$ conditioned by $y_t^m$, *i.e.*, $\hat{z}_t^m = h_a(y_t^m; \theta_{h_a})$, is a deterministic output without distribution property. Although it is feasible to approximate the $\hat{z}_t^m$ as the mean $\hat{\mu}_t^m$ of a Gaussian distribution (as the assumption of posterior distribution) while constructing a variance branch from scratch (Vahdat & Kautz, 2020), the optimization of the variance model may be difficult when model parameters are fixed during latent refinement phase. To address these issues, we utilize the dropout variational inference (Gal & Ghahramani, 2016) as a Bayesian approximation to existing hyperprior models for modeling the posterior distribution. Specifically, without introducing any new network, the deterministic

pre-trained model $\theta_{h_a}$ can be formulated as a probabilistic one $p(\theta_{h_a})$, by treating the weights as distributions. Thus, the estimated posterior estimation can be represented as follows,

$$p(z_t^m|y_t^m) \approx \hat{p}(z_t^m|y_t^m) = \mathcal{N}(\hat{\mu}_t^m, (\hat{\sigma}_t^m)^2), \tag{19}$$

where the posterior distribution is estimated and assumed as a fully factorized Gaussian distribution. The estimated mean $\hat{\mu}_t^m$ and variance $(\hat{\sigma}_t^m)^2$ can be represented as follows,

$$\hat{\mu}_t^m = \frac{1}{T}\sum_{i=1}^{T} h_a(y_t^m; \theta_{h_a}^i), \hat{\sigma}_t^m = \frac{1}{T}\sum_{i=1}^{T}[h_a(y_t^m; \theta_{h_a}^i) - \frac{1}{T}\sum_{i=1}^{T} h_a(y_t^m; \theta_{h_a}^i)]^2, \theta_{h_a}^i \sim q_\vartheta(\theta_{h_a}) \tag{20}$$

For the practical dropout variational inference (DVI) (Gal & Ghahramani, 2016), the dropout strategy (Srivastava et al., 2014) is conducted to render approximate samples from the posterior distribution, which equals to use a Bernoulli variational distribution $q_\vartheta(\theta_{h_a})$, parameterized by $\vartheta$, to approximate the true model weight posterior $p(\theta_{h_a})$. By conducting $T$ times of Monte Carlo (MC) sampling from $q_\vartheta(\theta_{h_a})$, we can estimate the mean and the variance of the posterior distribution $p(z_t^m|y_t^m)$. The dropout probability related to the Bernoulli variational distribution is set to 0.5. The computation consumption of the DVI is mild, as a single instance can parallelly sample $T$ masked weights by one inference in a batch of repeated instances. Moreover, it is flexible to degenerate to vanilla deterministic networks when the dropout probability is 1. Finally, we compute the negative log-likelihoods of the estimated posterior distribution given the current latent variable $z_t^m$.

***Discussion** – connection with Bit-back coding (BBC).* Both BBC (Townsend et al., 2019; Ruan et al., 2021; Ho et al., 2019) and our method derive from the joint probability approximation of marginal distribution. However, BBC usually specializes in *in-domain* image compression to narrow the marginalization gap, *i.e.,* transforming learned optimal joint probability to true marginal probability at compression time by minimizing the following objective, $\mathcal{L}_{BBC} =$

$$\mathbb{E}_{x_t \sim p_{x_t}, y_t, z_t \sim p_{y_t, z_t}}[-\log p(y_t^m|z_t^m, \theta_{h_e}^s) - \log p(z_t^m|\varphi^s) - (-\log p(z_t^m|y_t^m)) + \lambda(-\log p(x_t|y_t^m))]. \tag{21}$$

For cross-domain scenarios, such optimal joint probability does not hold due to mismatched encoding distribution as discussed in Eqs. (13) and (16). Instead, we focus on refining deteriorated joint probability to potentially optimal (even initial) one by minimizing the extra rate consumption of marginal approximation, as discussed in Eq. (18). More differences refer to Appendix.

## 4 EXPERIMENTS

**Datasets.** By following previous literature (Lv et al., 2023; Tsubota et al., 2023; Shen et al., 2023), we collect six different datasets with four types of image styles to comprehensively evaluate the R-D performance of different approaches on cross-domain TTA-IC tasks, including natural image (Kodak), screen content image (SIQAD Yang et al. (2015), SCID (Ni et al., 2017), CCT (Min et al., 2017)), pixel-style gaming image (Lv et al. (2023)' self-collected), and painting image (DomainNet (Peng et al., 2019)) datasets. The details of the used dataset can be found in the Appendix. Specifically, we consider the natural image dataset as in-domain evaluations, and others as cross-domain evaluations.

**Implementation Details.** We use CompressAI (Bégaint et al., 2020) to implement our proposed and baseline methods. Two widely-used backbone models are adopted, including the base hyperprior-based entropy model proposed by Ballé et al. (2018), namely Hyperprior, and the autoregressive context-based entropy model adopted by Cheng et al. (2020), namely AR-CM. Both are trained on natural images with various $\lambda$ settings. We use the pre-trained models provided by CompressAI. For TTA-IC, we use the same value of hyperparameter $\lambda = [0.0018, 0.0035, 0.0067, 0.013, 0.025, 0.048]$ for latent refinement. The Adam optimizer is utilized to update the latent variables in a learning rate of $1 \times 10^{-3}$ with 2000 iterations. $T$ is empirically set to 20 for MC sampling. We discuss different implementations and hyperparameter settings (*e.g.,* $\beta$) of dropout variational inference in sec. 4.3.

**Baselines.** For latent refinement methods, we adopt two common baselines: (i) BLR (Campos et al., 2019), as formulated in Eq. (14). (ii) HLR (Yang et al., 2020), as formulated in Eq. (15). For HLR, we follow Yang et al. (2020) to use a temperature annealing schedule with defaulted hyperparameters, where $\tau_0 = 0.5$, $c_0 = 0.001$, and $t_0 = 700$.

**Plug-and-play Validation.** We also replace different latent refinement methods of existing SOTA TTA-IC approaches (Tsubota et al., 2023; Shen et al., 2023) with our proposed latent refinement counterpart in a plug-and-play manner. We follow their defaulted hyperparameter settings.

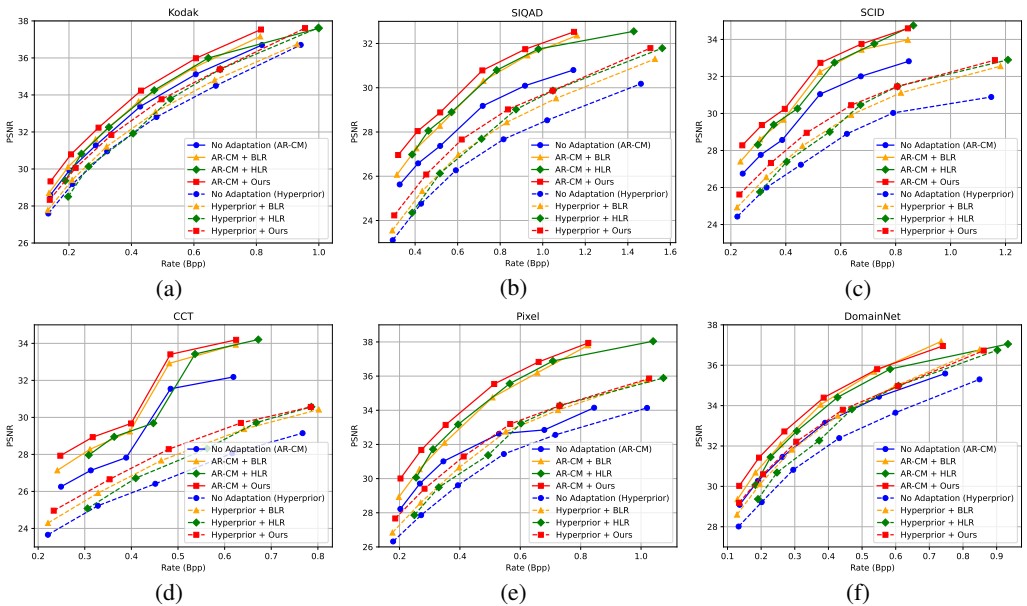

Figure 3: R-D curves on six datasets using different latent refinement methods. Two different base model architectures including AR-CM and Hyperprior are used.

Table 1: Comparison of our method with existing latent refinement approaches. The evaluation is measured in terms of average BD-rate savings (% ↓) using the respective base models (without test-time adaptation). Smaller values indicate superior performance.

| Method | In-domain dataset | Cross-domain datasets | | | | | |
|---|---|---|---|---|---|---|---|
| | Kodak | SIQAD | SCID | CCT | Pixel | DomainNet | Average↓ |
| AR-CM (Cheng et al., 2020) | 0 | 0 | 0 | 0 | 0 | 0 | 0 |
| + BLR | -6.75 | -18.75 | -18.31 | -25.33 | -26.22 | -18.78 | -21.41 |
| + HLR | -12.73 | -21.28 | -15.11 | -23.08 | -31.76 | -15.61 | -21.36 |
| + Ours | **-16.11** | **-24.71** | **-22.63** | **-28.26** | **-35.89** | **-23.52** | **-27.00** |
| Hyperprior (Ballé et al., 2018) | 0 | 0 | 0 | 0 | 0 | 0 | 0 |
| + BLR | -6.59 | -13.16 | -16.71 | -18.91 | -15.12 | -22.99 | -15.41 |
| + HLR | -11.32 | -13.94 | -14.59 | -13.82 | -18.89 | -16.68 | -14.87 |
| + Ours | **-14.82** | **-20.95** | **-23.51** | **-23.58** | **-22.22** | **-23.32** | **-20.78** |

Table 2: Integration of our method with the SOTA TTA-IC approaches. The evaluation is measured in terms of average BD-rate savings (% ↓) using the respective base models (without test-time adaptation). Smaller values indicate superior performance. ‡ Matrix decomposition-based adaptor. ⋆ Entropy efficient adapter. Stages 1 and 2 denote the latent refinement and decoder adaptation phases.

| Method | Type | | ID | In-domain dataset | Cross-domain datasets | | | | | Average↓ |
|---|---|---|---|---|---|---|---|---|---|---|
| | | | | Kodak | SIQAD | SCID | CCT | Pixel | DomainNet | |
| Tsubota et al. (2023) | WACNN (Zou et al., 2022) | | | 0 | 0 | 0 | 0 | 0 | 0 | 0 |
| | Stage 1 | + HLR† | (a) | -3.50 | -13.20 | -12.34 | -17.15 | -11.08 | -2.94 | -10.02 |
| | | + Ours | (b) | **-5.45** | **-14.32** | **-14.62** | **-18.95** | **-13.01** | **-5.46** | **-12.01** |
| | Stage 2 | (a) + Adaptor1‡ | (c) | -3.51 | -20.98 | -20.73 | -24.90 | -17.23 | -2.09 | -14.89 |
| | | (b) +Adaptor1‡ | (d) | **-5.48** | **-22.70** | **-22.65** | **-26.60** | **-19.37** | **-4.51** | **-16.95** |
| Shen et al. (2023) | WACNN (Zou et al., 2022) | | | 0 | 0 | 0 | 0 | 0 | 0 | 0 |
| | Stage 1 | + BLR | (e) | 3.83 | -10.98 | -9.96 | -13.05 | -3.13 | 4.83 | -4.74 |
| | | + Ours | (f) | **-5.45** | **-14.32** | **-14.62** | **-18.95** | **-13.01** | **-5.46** | **-12.01** |
| | Stage 2 | (e) + Adaptor2⋆ | (g) | 2.63 | -18.91 | -18.74 | -22.17 | -11.88 | 3.41 | -10.94 |
| | | (f) + Adaptor2⋆ | (h) | **-6.01** | **-20.23** | **-21.02** | **-24.23** | **-14.99** | **-5.59** | **-15.26** |

## 4.1 COMPARISON WITH LATENT REFINEMENT METHODS

**R-D Performance.** We compare our proposed method with BLR and HLR. Note that the model without adaptation is also involved, namely No Adaptation. The peak signal-to-noise ratio (PSNR) and the data rate in bits per pixel (bpp) on different quality levels (refers to different $\lambda$ levels) are calculated to evaluate the R-D performance of different methods. Then, the R-D curves can be plotted. The results can be found in Figure 3. We can observe that all latent refinement methods achieve comparable performance, which may be reasonable as the Kodak dataset can be regarded as the

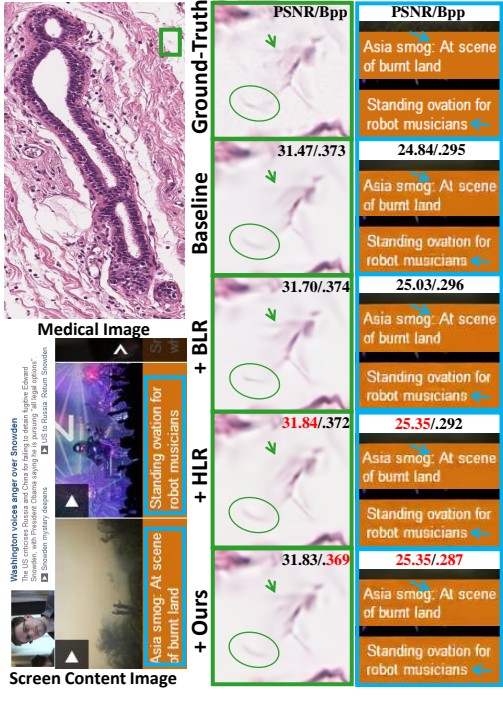

Figure 4: Entropy curves of different probabilities with the iteration $m$, including (a) $-\log p(y_t^m)$ (b)$-\log p(z_t^m)$ (c) $-\log p(z_t^m|y_t^m)$ (d) validation loss as calculated in Eq. (3).

Figure 6: Qualitative results

Figure 7: R-D performance on medical images, where pathological breast cancer images are used for cross-domain compression.

Table 3: Ablation study of proposed distribution regularization $\mathcal{L}_{DR}$ in terms of BD-rate (% ↓) on SIQAD. DL denotes the number of the dropout layer. †: assume $\sigma = 1$ without Bayesian approximation. ‡: minimizing Eq. (21).

| Baseline (HLR) | 0: As an anchor | | |
|---|---|---|---|
| | DL \ $\beta$ | 0.1 | 1 | 10 |
| + proposed $\mathcal{L}_{DR}$ | 1 | -10.01 | -10.02 | -7.07 |
| | 2 | -10.03 | -9.62 | -5.25 |
| | 3 | **-10.06** | -9.61 | -4.95 |
| + Deterministic $\mathcal{L}_{DR}^{\dagger}$ | -3.54 | | |
| + BBC (Yang et al., 2020)‡ | 0.15 | | |

in-domain data without domain shifts. The gains of HLR and our proposed method mainly derive from the minimization of the discretization gap compared with BLR. For out-of-domain images such as SIQAD, SCID, CCT, Pixel, and DomainNet, our proposed method outperforms other approaches with a clear margin regardless of different backbones.

**Bjøntegaard Delta bit-rate (BD-rate)**. We compute the BD-rate (Bjøntegaard, 2001) for performance evaluation. A lower BD rate demonstrates better performance. The baseline model without adaptation is the anchor model for BD-rate calculation. The results are shown in Table 1. First, the HLR and our proposed method outperform the BLR with a clear margin for the in-domain Kodak dataset, which coincides with Yang et al. (2020)' obervations. However, on the cross-domain tasks, the HLR suffers from significant degradations compared with BLR, where HLR performs below the HLR on three out of five tasks, *e.g.,* SCID and CCT. This is reasonable, as the static entropy bottleneck correlated with the distribution property of natural images cannot model the true distribution of latent variable $z$ well. Finally, our method can achieve obvious gains on all cross-domain tasks.

## 4.2 PLUGGING INTO SOTA TTA-IC

Moreover, we replace the existing latent refinement methods used by SOTA TTA-IC approaches with our proposed method. The full R-D curves are in the Appendix. We present the BD-rate in Table 2. It is obvious that our proposed method can improve the performance of these TTA-IC methods on all tasks. Moreover, such gains are roughly maintained at a similar level when adopted by the decoder adaptation. Due to the benefit of hybrid latent refinement, these gains are significantly enlarged for Shen et al. (2023). Thus, our proposed method enjoys plug-and-play property, benefiting existing TTA-IC methods.

### 4.3 IN-DEPTH ANALYSES OF OUR PROPOSED METHOD

**Why does the proposed distribution regularization improve the R-D performance? (i)** From theoretical analyses in section 3.2, the domain shifts and the underlying mismatch between $p(z_t^m|\varphi^s)$ and $p(y_t^m|z_t^m, \theta_{h_e}^s)$ may trigger the deteriorated joint probability approximation $p(y_t^M, z_t^M)$ of true marginal distribution $p(y_t^*)$ for vanilla HLR. By empirical experiments in Figure 4, we observe that it is indeed difficult to jointly optimize latent variable

Table 4: Correlation between adaptation performance and adaptation time (using a single NVIDIA GeForce 3090 GPU) on SIQAD.

| | Steps | 1 | 500 | 1000 | 1500 | 2000 |
|---|---|---|---|---|---|---|
| **Runtime (Avg. Sec./Img.)** | BLR | 0.07s | 13.96s | 27.32s | 41.52s | 56.28s |
| | HLR | 0.07s | 14.86s | 28.52s | 42.51s | 57.58s |
| | Ours | 0.08s | 15.48s | 29.25s | 44.85s | 59.70s |
| **BD-rate(%)↓** | BLR | **-1.01** | -19.23 | -20.57 | -21.04 | -21.27 |
| | HLR | -0.92 | -12.75 | -14.73 | -19.32 | -23.68 |
| | Ours | -0.95 | **-24.33** | **-29.88** | **-33.15** | **-34.20** |

$y_t$ and side information $z_t$ for vanilla HLR as shown in Figs. 4(a) and 4(b), where $-\log p(z_t^m)$ converges first and degrades with the iterations while $-\log p(y_t^m)$ converges well. In contrast, due to the lack of optimization for slide information $z_t$, there is limited convergence of $-\log p(y_t^m)$ for the BLR while $-\log p(z_t^m)$ is unchanged. Thus, as shown in Figure 4(c), there is an obvious degradation for vanilla HLR, *i.e.,* $\Delta H^0 < \Delta H^M$, reflecting more rate consumption of marginal approximation. This implies that the finally updated joint probability $p(y_t^M, z_t^M)$ is further away from the optimal joint probability $p(y_t^*, z_t^*)$, and even the initial joint probability $p(y_t^0, z_t^0)$. **(ii)** By the proposed distribution regularization (refers to Figure 4(c)), the degradation of joint probability approximation is alleviated well (blue curve) compared with vanilla HLR. Even, our method can achieve good convergence, *i.e.,* $\Delta H^0 > \Delta H^M$, which is reasonable as we directly minimize $\Delta H$ as shown in Eq. (18).
*In short, the distribution regularization encourages the deteriorated joint probability approximation to approach the initial and even unknown optimal ones with a good convergence of rate consumption.*

**How about the effectiveness of Bayesian approximation for distribution regularization?** First, we ablate the number of DVI layers. Intuitively, more dropout layers facilitate a more accurate estimate of the posterior distribution. As shown in Table 3, there are marginal gains with improving dropout layers. Moreover, over-regularization, *i.e.,* $\beta = 10$, will result in obviously negative effects. Thus, to avoid performance degradation, it is better to construct more dropout layers with more accurate probability representation and set a lower regularization coefficient (such as 0.1) to alleviate the over-regularized downside. Second, we try a deterministic version of our distribution regularization by removing the Bayesian approximation, *i.e.,* assuming $\sigma = 1$ without the dropout layer. As we can see, deterministic $\mathcal{L}_{DR}$ performs significantly below that of the Bayesian approximation-based one, which is reasonable as the over- or below-estimated variance of the posterior distribution is inaccurate compared with the practical DVI in MC sampling. Finally, we observe negligible gains of the BBC on cross-domain image compression, which coincides with our discussions in section 3.3.

**How about the adaptation efficiency compared with baseline methods?** As illustrated in Table 4, our proposed method has obvious performance gains in different adaptation steps. In contrast, the additional adaptation time taken by ours is mild in the same adaptation step which is reasonable as an instance can parallelly sample $T$ masked weights by one inference in a batch of repeated instances.

**What is the ability to generalize for larger distribution discrepancy?** We explore whether our proposed method can be scalable to more challenging medical images, where the image distribution of medical images is quite different from natural images. The qualitative and quantitative results in Figures. 6 and 7 show our method can improve R-D performance with better texture preservation.

## 5 LIMITATION AND CONCLUSION

**Limitation.** Due to the introduction of the DVI, our proposed method has a minor increase in adaptation cost compared with baseline methods, as illustrated in Table 4. developing a more efficient solution and using a more powerful GPU on the server can reduce the adaptation time in the future.

We approach an advanced latent refinement method by tailoring the vanilla HLR designed for *in-domain* inference improvement to *cross-domain* cases. Specifically, we provide a theoretical analysis to uncover that the underlying mismatch between refined Gaussian conditional and hyperprior distributions may trigger the deteriorated joint probability approximation of marginal distribution. Then, we introduce a Bayesian approximation-endowed *distribution regularization* to encourage learning better joint probability approximation in a plug-and-play manner. Extensive experiments demonstrate that our proposed method can achieve promising performance.

## ACKNOWLEDGMENTS

This work was supported in part by the Hong Kong Innovation and Technology Commission (ITC) (InnoHK Project CIMDA), in part by the Institute of Digital Medicine of City University of Hong Kong (Project 9229503), in part by the Hong Kong Research Grants Council under Projects 21200522, 11200323 and 11203220, in part by Chow Sang Sang Donation and Matching Fund (Project 9229161), and in part by the Hong Kong Innovation and Technology Commission (Project GHP/044/21SZ).

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

## A FULL DERIVATIONS

**Proposition 1.** *Let $y$ and $z$ be the latent and hyper latent variables, and these variables with the asterisk be their optimal representations. In the context of in-domain image compression, if an optimal joint probability approximation of true marginal distribution can be achieved by minimizing Eq. (3), the extra rate consumption of marginalization approximation is*

$$\Delta H^* = H(y, z) - H(y^*) = \mathbb{E}_{y,z \sim p(y,z)}[-\log p(z|y)], \tag{22}$$

*Proof.* On the one hand, with joint probability and the Bayesian rule $p(y^*) = \frac{p(y^*|z^*)p(z^*)}{p(z^*|y^*)}$, we have

$$H(y, z) = \mathbb{E}_{y,z \sim p(y|z)p(z)}[-\log p(y|z)] + \mathbb{E}_{z \sim p(z)}[-\log p(z)], \tag{23}$$

$$H(y^*) = \mathbb{E}_{y^*,z^* \sim p(y^*|z^*)p(z^*)}[-\log p(y^*|z^*)] \quad + \quad \mathbb{E}_{z^* \sim p(z^*)}[-\log p(z^*)] \\ - \quad \mathbb{E}_{y^*,z^* \sim p(y^*,z^*)}[-\log p(z^*|y^*)] \tag{24}$$

Then, we have

$$\begin{aligned} \Delta H^* &= [\mathbb{E}_{y,z \sim p(y|z)p(z)}[-\log p(y|z)] - \mathbb{E}_{y^*,z^* \sim p(y^*|z^*)p(z^*)}[(-\log p(y^*|z^*))]] \\ &+ [\mathbb{E}_{z \sim p(z)}[-\log p(z)] - \mathbb{E}_{z^* \sim p(z^*)}[(-\log p(z^*))]] \\ &+ \mathbb{E}_{y^*,z^* \sim p(y^*,z^*)}[-\log p(z^*|y^*)]. \end{aligned} \tag{25}$$

On the other hand, as there exist optimal probability representations for $p(y|z)$ and $p(z)$ for in-domain image compression by minimizing Eq. (3), we have

$$\mathbb{E}_{y,z \sim p(y|z)p(z)}[-\log p(y|z)] - \mathbb{E}_{y^*,z^* \sim p(y^*|z^*)p(z^*)}[(-\log p(y^*|z^*))] = 0, \\ s.t. \quad p(y|z) = p(y^*|z^*) \quad \text{and} \quad p(z) = p(z^*), \tag{26}$$

$$\mathbb{E}_{z \sim p(z)}[-\log p(z)] - \mathbb{E}_{z^* \sim p(z^*)}[(-\log p(z^*))] = 0, \quad s.t. \quad p(z) = p(z^*) \tag{27}$$

Thus,

$$\Delta H^* = \mathbb{E}_{y^*,z^* \sim p(y^*,z^*)}[-\log p(y^*|z^*)] = \mathbb{E}_{y,z \sim p(y,z)}[-\log p(z|y)] \tag{28}$$

For in-domain image compression, Eq. (26), Eq. (27), and Eq. (28) hold, as $p(y|z)$, $p(z)$, and $p(z|y)$ are close to optimal probability representations $p(y^*|z^*)$, $p(z^*)$, and $p(z^*|y^*)$ due to the assumption of the optimal joint probability approximation of true marginal distribution. $\square$

**Corollary 1.** *The extra rate consumption of cross-domain marginalization approximation $\Delta H$ will be larger than that of in-domain marginalization approximation, i.e,, $\Delta H > \Delta H^*$, due to deteriorated joint probability.*

*Proof.* Due to distribution shifts and suboptimal source-domain parameters $(\theta_{h_e}^s, \varphi^s)$, Eqs. (26) and (27) in Prop. 1 can be further represented based on Prop. 2 as follows,

$$\mathbb{E}_{y_t,z_t \sim p(y_t|z_t,\theta_{h_e}^s)p(z_t)}[-\log p(y_t|z_t,\theta_{h_e}^s) - \mathbb{E}_{y_t^*,z_t^* \sim p(y_t^*|z_t^*))p(z_t^*)}[(-\log p(y_t^*|z_t^*))] > 0, \\ s.t. \quad p(y_t|z_t,\theta_{h_e}^s) \neq p(y_t^*|z_t^*) \text{ and } p(z_t|\varphi^s) \neq p(z_t^*), \tag{29}$$

$$\mathbb{E}_{z_t \sim p(z_t|\varphi^s)}[-\log p(z_t|\varphi^s)] - \mathbb{E}_{z_t \sim p(z_t^*)}[(-\log p(z_t^*)] > 0, \quad s.t. \quad p(z_t|\varphi^s) \neq p(z_t^*), \tag{30}$$

where $p(y_t|z_t,\theta_{h_e}^s) \neq p(y_t^*|z_t^*)$ holds due to source image-correlated $\theta_{h_e}^s$ and unreliable $z_t$. $p(z_t|\varphi^s) \neq p(z_t^*)$ holds due to poor specialization of entropy bottleneck $\varphi^s$. Let $p(y_t|z_t,\theta_{h_e}^s)=p(y_t|z_t)$ and $p(z_t|\varphi^s)=p(z_t)$ for brevity. We have

$$\begin{aligned} \Delta H &= \mathbb{E}_{y,z \sim p(y|z)p(z)}[-\log p(y_t|z_t)] - \mathbb{E}_{y^*,z^* \sim p(y^*|z^*)p(z^*)}[(-\log p(y_t^*|z_t^*))] \\ &+ \mathbb{E}_{z \sim p(z)}[-\log p(z)] - \mathbb{E}_{z^* \sim p(z^*)}[(-\log p(z^*))] \\ &+ \mathbb{E}_{y^*,z^* \sim p(y^*,z^*)}[-\log p(z_t^*|y_t^*)] > \Delta H^* \end{aligned} \tag{31}$$

Since optimal probabilities always match the true underlying distributions from Prop. 2, $p(z_t^*|y_t^*)$ is equivalent to $p(z^*|y^*)$ in Eq. 28 for optimal posterior distributions. Thus, Eq.31 holds and implies more rate consumption is potentially incurred in cross-domain scenarios. $\square$

Here, we further explain why the Eqs. (29) and (30) hold as follows.

According to Shannon's entropy theorem, the optimal probability distribution $p(y_t^*|z_t^*)$ achieves the theoretical minimum coding length, which equals the entropy of the true data distribution. Any non-optimal coding distribution (e.g., $p(y_t|z_t, \theta_{h_e}^s)$) must have a higher expected code length than the theoretical minimum. The entropy consumption of coding distribution can reflect this to make the following inequality strictly hold:

$$\mathbb{E}_{y,z \sim p(y_t|z_t)p(z_t)}[-\log p(y_t|z_t, \theta_{h_e}^s)] \geq \mathbb{E}_{y^*,z^* \sim p(y_t^*|z_t^*)p(z_t^*)}[-\log p(y_t^*|z_t^*)], \qquad (32)$$

where the equality holds only if $p(y_t^*|z_t^*) = p(y_t|z_t, \theta_{h_e}^s)$ and $p(z_t|\varphi^s) = p(z_t^*)$.

Since we have assumed that $p(y_t|z_t, \theta_{h_e}^s) \neq p(y_t^*|z_t^*)$ holds due to source image-correlated $\theta_{h_e}^s$ and unreliable $p(z_t|\varphi^s) \neq p(z_t^*)$ on cross-domain scenarios. Thus,

$$\mathbb{E}_{y,z \sim p(y_t|z_t)p(z_t)}[-\log p(y_t|z_t, \theta_{h_e}^s)] > \mathbb{E}_{y^*,z^* \sim p(y_t^*|z_t^*)p(z_t^*)}[-\log p(y_t^*|z_t^*)] \qquad (33)$$

holds. Thus, the following inequality can be naturally derived,

$$\mathbb{E}_{y,z \sim p(y_t|z_t)p(z_t)}[-\log p(y_t|z_t, \theta_{h_e}^s)] - \mathbb{E}_{y^*,z^* \sim p(y_t^*|z_t^*)p(z_t^*)}[-\log p(y_t^*|z_t^*)] > 0. \qquad (34)$$

To summarize, it is clear Eq. (29) holds. The same applies to Eq. (30).

## B    DETAILS OF ADOPTED DATASETS

By following previous literature (Lv et al., 2023; Tsubota et al., 2023; Shen et al., 2023), we collect six different datasets with four types of image styles to comprehensively evaluate the R-D performance of different approaches on cross-domain TTA-IC tasks, including natural image (Kodak[1]), screen content image (SIQAD Yang et al. (2015), SCID (Ni et al., 2017), CCT (Min et al., 2017)), pixel-style gaming image (Lv et al. (2023)' self-collected), and painting image (DomainNet (Peng et al., 2019)) datasets. The details of the used dataset can be found in the Table 5. Specifically, we consider the natural image dataset as in-domain evaluations, and others as cross-domain evaluations.

## C    COMPARISON WITH SOTA TTA-IC METHODS

We provide the full R-D curves in Figure 8 when we compared our proposed method with SOTA TTA-IC methods. Note that our proposed method is based on (Tsubota et al., 2023) with our proposed distribution regularization. We can observe that all latent refinement methods achieve comparable performance, which may be reasonable as the Kodak dataset can be regarded as the in-domain data without domain shifts. The gains of HLR and our proposed method mainly derive from the minimization of the discretization gap compared with BLR. For out-of-domain images such as SIQAD, SCID, CCT, Pixel, and DomainNet, our proposed method outperforms other approaches with a clear margin regardless of different backbones. Especially, a better R-D performance on the low-bit conditions can be observed for AR-CM-based realizations on the Pixel dataset.

## D    CONNECTION WITH BIT-BACK CODING

1) *In-domain v.s. Cross-domain*: Both BBC (Townsend et al., 2019; Ruan et al., 2021; Ho et al., 2019) and our proposed method derive from the joint probability approximation of marginal distribution. However, BBC usually specializes in *in-domain* image compression to narrow the marginalization gap, *i.e.,* transforming learned optimal joint probability to true marginal probability at compression time. Instead, in the context of *cross-domain* image compression, such optimal joint probability does

---

[1]https://r0k.us/graphics/kodak/

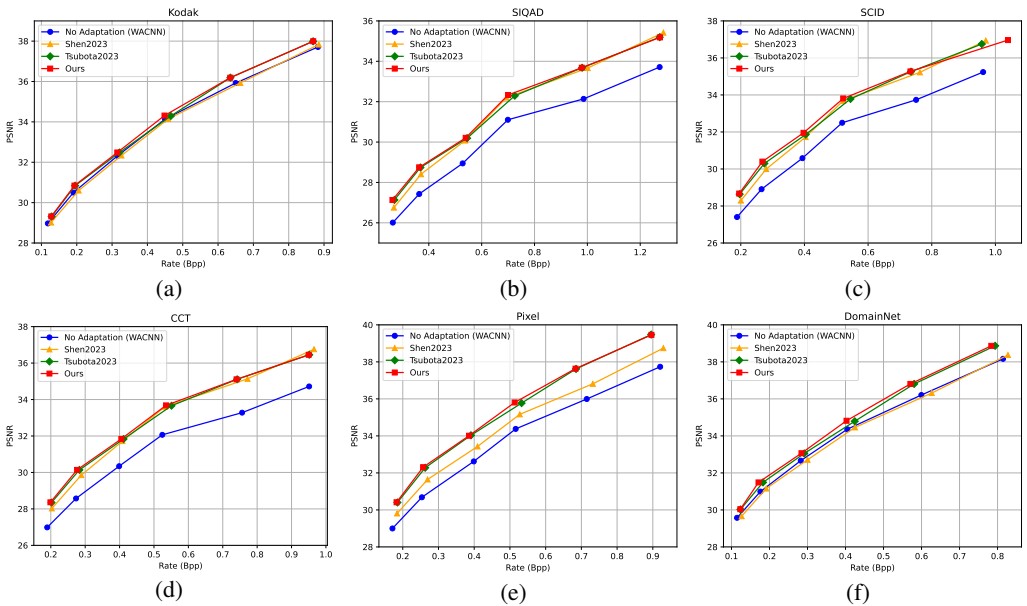

Figure 8: R-D curves of test-time adaptation of different latent refinement approaches on six datasets using different latent refinement methods. Two different base model architectures including AR-CM and Hyperprior are used.

Table 5: The datasets for evaluation. The symbols $*$ and $\dagger$ denote in- and cross-domain datasets, respectively.

| Dataset | Description | # Num. | Avg. Resolution |
|---|---|---|---|
| Kodak$^*$ | Natural | 24 | $576{\times}704$ |
| SIQAD$^\dagger$ | Screen Content | 24 | $685{\times}739$ |
| SCID$^\dagger$ | Screen Content | 40 | $720{\times}1080$ |
| CCT$^\dagger$ | Screen Content | 24 | $915{\times}1627$ |
| Pixel$^\dagger$ | Gaming | 25 | $746{\times}850$ |
| DomainNet$^\dagger$ | Painting | 25 | $492{\times}640$ |

not hold due to mismatched encoding distribution (*e.g.,* entropy bottleneck), as discussed in Eqs. (13) and (16).

2) *Minimizing posterior probability v.s. Maximizing posterior probability*: BBC minimizes the second term of the first line of Eq. (22) as the true entropy of the marginal distribution, *i.e.,*

$$\mathcal{L}_{BBC} = -\log p(y_t^m | z_t^m, \theta_{h_e}^s) - \log p(z_t^m | \varphi^s) - (-\log p(z_t^m | y_t^m)) + \lambda(-\log p(x_t | y_t^m)), \quad (35)$$

where the third term corresponds to minimizing the posterior probability based on the in-domain optimal joint probability assumption of BBC. In contrast, our proposed method focuses on refining deteriorated joint probability to optimal one by minimizing the extra rate consumption of marginal approximation, as discussed in Eq. (18).

3) *Performance and implementation differences*: Our experiments in Table 3 observe negligible gains of BBC on cross-domain image compression. More importantly, in order to acquire distribution property, our proposed distribution regularization relies on more flexible variational Bayesian inference rather than introducing additional networks like BBC (Yang et al., 2020).

# E   MORE PLUG-AND-PLAY VALIDATIONS

To further the effectiveness of this plug-and-play advantage, we have conducted additional experiments on more baseline methods. The results can be observed in Table 6. As we can see, our proposed method can improve the performance of these TTA-IC methods in a plug-and-play manner, which is

Table 6: Integration of our method with the SOTA TTA-IC approaches. The evaluation is measured in terms of average BD-rate savings (% ↓) using the respective base models (without test-time adaptation). Smaller values indicate superior performance. ‡ Matrix decomposition-based adaptor. ⋆ Entropy efficient adapter. † dynamic low-rank adaptor. ‡‡: Only latent representation $y$ is optimized. Stages 1 and 2 denote the latent refinement and decoder adaptation phases.

| Method | | Type | ID | In-domain dataset Kodak | Cross-domain datasets SIQAD | SCID | CCT | Pixel | DomainNet | Average↓ |
|---|---|---|---|---|---|---|---|---|---|---|
| Tsubota et al. (2023) | | WACNN (Zou et al., 2022) | | 0 | 0 | 0 | 0 | 0 | 0 | 0 |
| | Stage 1 | + HLR | (a) | -3.50 | -13.20 | -12.34 | -17.15 | -11.08 | -2.94 | -10.02 |
| | | + Ours | (b) | **-5.45** | **-14.32** | **-14.62** | **-18.95** | **-13.01** | **-5.46** | **-12.01** |
| | Stage 2 | (a) + Adaptor1‡ | (c) | -3.51 | -20.98 | -20.73 | -24.90 | -17.23 | -2.09 | -14.89 |
| | | (b) +Adaptor1‡ | (d) | **-5.48** | **-22.70** | **-22.65** | **-26.60** | **-19.37** | **-4.51** | **-16.95** |
| Shen et al. (2023) | | WACNN (Zou et al., 2022) | | 0 | 0 | 0 | 0 | 0 | 0 | 0 |
| | Stage 1 | + BLR | (e) | 3.83 | -10.98 | -9.96 | -13.05 | -3.13 | 4.83 | -4.74 |
| | | + Ours | (f) | **-5.45** | **-14.32** | **-14.62** | **-18.95** | **-13.01** | **-5.46** | **-12.01** |
| | Stage 2 | (e) + Adaptor2⋆ | (g) | 2.63 | -18.91 | -18.74 | -22.17 | -11.88 | 3.41 | -10.94 |
| | | (f) + Adaptor2⋆ | (h) | **-6.01** | **-20.23** | **-21.02** | **-24.23** | **-14.99** | **-5.59** | **-15.26** |
| Lv et al. (2023) | | WACNN (Zou et al., 2022) | | 0 | 0 | 0 | 0 | 0 | 0 | 0 |
| | Stage 1 | + HLR‡‡ | (i) | -2.25 | -11.49 | -10.26 | -14.89 | -7.87 | -1.58 | -9.66 |
| | | + Ours | (j) | **-5.45** | **-14.32** | **-14.62** | **-18.95** | **-13.01** | **-5.46** | **-12.01** |
| | Stage 2 | (i) + Adaptor3† | (k) | -3.45 | -16.78 | -17.05 | -22.89 | -18.01 | -3.28 | -13.57 |
| | | (j) + Adaptor3† | (m) | **-5.89** | **-18.59** | **-19.89** | **-25.68** | **-20.12** | **-6.21** | **-16.06** |

Table 7: Comparison of our method with existing latent refinement approaches. The evaluation is measured in terms of average BD-rate savings (% ↓) using the base models (without test-time adaptation). Smaller values indicate superior performance.

| Method | In-domain datasets Set5 | Set14 | BSD100 | Urban100 | Average↓ |
|---|---|---|---|---|---|
| AR-CM (Cheng et al., 2020) | 0 | 0 | 0 | 0 | 0 |
| + BLR | -13.27 | -15.32 | -5.92 | -16.25 | -12.69 |
| + HLR | -15.01 | -18.29 | -6.61 | -21.42 | -15.33 |
| + Ours | **-21.47** | **-24.88** | **-11.23** | **-26.68** | **-21.06** |

reasonable as our proposed method can alleviate deteriorated joint probability approximation, leading to a better bitrate budget in a two-stage TTA-IC framework.

# F   MORE VALIDATIONS ON POPULAR DATASETS

We utilize Set5, Set14, BSD100, and Urban100 benchmark datasets for further evaluations, where Set 5 and Set14 include 5 and 14 images, respectively, both BSD100 and Urban100 have 100 high-resolution images. As illustrated in Table 7, our proposed method can surpass baseline methods in terms of the BD-rate. Such performance gains show a good generalization ability of our proposed method on diverse datasets, exhibiting its usability in the real world.

# G   ADAPTATION COST

We compute the adaptation cost of different methods in terms of GPU memory usage and GFLOPs. As illustrated in Table 8, our proposed method has a minor increase in adaptation cost, which is reasonable as the dropout variational inference is introduced during optimization for better R-D performance.

Notably, although our proposed method increases the adaptation cost, such a latent refinement stage is usually conducted on the encoding side of the server, which means that there is no effect on the real-time decoding on edge devices, making our proposed method applicable to real-world scenarios. Moreover, our proposed method has no model modification to obtain probability representation based on dropout variational inference. This simplification also enhances its usability.

Table 8: Average adaptation cost of different methods on Kodak and CCT datasets.

|  | BLR | HLR | Ours |
|---|---|---|---|
| **GPU Memory (GB)** | 0.72/3.67 | 0.74/3.92 | 0.79/3.98 |
| **GFLOPs** | 98.98/525.81 | 101.23/563.24 | 110.58/587.47 |

## H   MORE DISCUSSION ABOUT LEARNING MATCHED POSTERIOR DISTRIBUTION

Here, we discuss a potential method, namely direct inferring, which conducts a directly inferring through a hyper analysis transform to address the mismatched problem. Although it is feasible to conduct a direct inferring through a hyper analysis transform in out-of-distribution (OOD) scenarios, we observe that the result is significantly behind the proposed distribution regularization. For example, by using the ablation study in Table 3, the result achieved by direct inferring is -4.84, which seems to be similar to the setting with DL=3 and $\beta = 10$ (-4.95) and the deterministic distribution regularization (-3.54).

We conjecture that the direct inferring may be a deterministic over-regularization method. Since the optimization of latent representation is directly affected by the reconstruction loss and rate cost of latent representation, the usage of direct inferring result $z$ still cannot be ensured to lead to minimal rate cost of side information and appropriate conditional distribution $p(y|z)$. Moreover, the optimization of side information $z$ uses the direct inferring result may result in a potential optimization difficulty, due to continually variational $z$ obtained by a hyper analysis transform.

Therefore, it seems that an appropriate regularization like the proposed distribution regularization is necessary to alleviate the optimization difficulty between latent representation and side information, leading to more matched joint probability.

