# OpenReview forum: "Test-time Adaptation for Image Compression with Distribution Regularization"
_ICLR.cc/2025/Conference — ICLR 2025 Poster_

### Official Review · Reviewer_5d5v · 2024-10-30

**Soundness:** 3
**Presentation:** 3
**Contribution:** 2
**Rating:** 6
**Confidence:** 1

**Summary:**

The author propose a novel test-time adaptation framework for image compression through directly refine the latent variables without altering any model parameters.

**Strengths:**

This research topic of this work seems a bit a minority (image compression + test-time adaptation) and contains a lot of background knowledge. After hard attempts, I think I still can't give an accurate evaluation. It is recommended that AC find another highly relevant reviewer in order to make a more comprehensive judgment.

**Weaknesses:**

None

**Questions:**

None

---

> ### Author Response · Authors · 2024-11-19
> **Thank you**
>
> Dear Reviewer,
>
> Thank you for dedicating your time and effort to reviewing our paper.
>
> **Comment:** This research topic of this work seems a bit a minority (image compression + test-time adaptation) and contains a lot of background knowledge.
>
> **Response:**
>
> **Test-time adaptation of image compression is necessary and important:** In learned image compression community, it is common that we encounter different styles of images (e.g., medical, screen content, and cartoon images) to be compressed. Due to the violation of
>  the i.i.d. assumption, it is challenging for advanced neural compressor to handle these out-of-the-distribution images, thus posing the emergence and importance of the test-time adaptation of image compression. Therefore, quite a few works are exploring promising test-time adaptation solution, such as latent refinement and decoder adaptation. Our proposed method belongs to the latent refinement method, which have played a basic role in SOTA TTA-IC models.
>
> **Our proposed method is a versatile solution and exhibits real-world applications:** Our method addresses a critical gap in cross-domain image compression, enhancing the rate-distortion performance, which is crucial for real-world applications like streaming and storage. By providing a theoretical analysis foundation and demonstrating practical improvements across diverse datasets, our approach not only advances the state-of-the-art but also offers a versatile solution that can be integrated into existing TTA-IC systems. This flexibility and practical impact underscore the broader relevance and potential of our work to drive future research and applications in the field.
>
> **Provide more background knowledge of image compression.** In the revised manuscript, we have added a preliminary section to provide more background knowledge of image compression to the reader. We hope the reader can understand this paper better. **Please refer to Page 3 for more detail.**

---

### Official Review · Reviewer_ShHq · 2024-11-01

**Soundness:** 2
**Presentation:** 2
**Contribution:** 2
**Rating:** 6
**Confidence:** 3

**Summary:**

This article investigates test-time adaption image compression method on cross-domain compression tasks from the perspective of latent refinement, by extending hybrid latent refinement and introducing a simple Bayesian approximation-endowed distribution regularization. The authors provide theoretical analysis and experimental results to demonstrate the performance of this method. My detailed comments are as follows.

**Strengths:**

1.	The authors propose their method from the perspective of latent refinement, which hasn’t been fully investigated. Compared with HLR, the method proposed by the author greatly improves the performance cross-domain compression tasks.
2.	The authors provide sufficient theoretical analysis on the method proposed in the article.

**Weaknesses:**

1.	The experimental results of this plug-and-play method are relatively weak, which cannot illustrate the generalization ability of this method on other TTA-IC approaches. It would be better to apply this method to more baseline approaches.
2.	It would be better to report the image compression results on top of popular image reconstruction benchmarks, such as Set5, Set14, BSD100, Urban100, etc.
3.	This paper is hard to follow and the writing could be further improved. It would be better to explicitly highlight the key contributions of this paper.
4.	The compared methods BLR and HLR are too old since both are published before 2021. It would be better to compare with more recent work, such as [A].
[A] Downscaled representation matters: Improving image rescaling with collaborative downscaled images, ICCV 2023.

**Questions:**

The compared methods are too old. More popular image reconstruction/compression datasets should be considered for comparisons.

---

> ### Author Response · Authors · 2024-11-19
> **Thank you; Address your concerns [1/2]**
>
> Dear Reviewer,
>
> Thank you for dedicating your time and effort to reviewing our paper. We are glad you think our method is meaningful, performing well, and providing sufficient theoretical analysis. We took your suggestions very carefully, and have conducted additional experiments to eliminate your concerns. These modifications can be found in the revised paper **(marked by the orange)**.
> ***
> **General Clarification:**
> - Your core concern is the baseline methods, such as BLR and HLR, are too old since both were published before 2021. **To the best of our knowledge, in the image compression community, these two latent refinement approaches play a fundamental role in recent state-of-the-art TTA-IC models (such as [B]-2023,[C]-2023, and [D]-2023), especially the HLR.** Note that existing SOTA TTA-IC methods use either the BLR or the HLR. **To the best of our knowledge, there are no new baselines beyond the BLR and HLR in neural image compression community. Thus, our comparison using the BLR and HLR is necessary and meaningful.**
>
> - To demonstrate the effectiveness of our proposed method on recent baselines, **we have integrated our proposed method into these recent TTA-IC baselines (such as [B]-2023,[C]-2023, and [D]-2023) for evaluation.** To conclude, compared with the emergence of various decoder refinement variants, the latent refinement is barely tailored to cross-domain scenarios, exhibiting underexplored space. This is one of our motivations in this paper.
>
> - **While we kindly appreciate your suggested baseline [A]-2023, this image rescaling work may not be directly applied to image compression due to various reasons (refer to point-by-point response).** In light of its valuable contributions (the first attempt to optimize the downscaled representation instead of the model), we will cite this paper to provide detailed discussions in the related work of the revised paper. Please find the point-by-point response as follows.
>
> ***
> ***
>
> **W1:** *The experimental results of this plug-and-play method are relatively weak, which cannot illustrate the generalization ability of this method on other TTA-IC approaches. It would be better to apply this method to more baseline approaches.*
>
> **Response:** Thanks for your comment.
>
> In Table 2, we have reported the experimental results when our proposed method is plugged into SOTA TTA-IC methods. To further the effectiveness of this plug-and-play advantage, we have conducted additional experiments on more baseline methods. The summarized results can be observed in the following table. As we can see, our proposed method can improve the performance of these TTA-IC methods in a plug-and-play manner, which is reasonable as our proposed method can alleviate deteriorated joint probability approximation, leading to a better bitrate budget in a two-stage TTA-IC framework.
>
> **Due to the length limit of the paper, please refer to the full Table 6 (Page 15) in the revised manuscript.**
>
> *Table 6 Integration of our method with SOTA TTA-IC approaches (BD-rate savings % ↓)*
> | Method  | Stage   | Type      | Average | Improvement |
> |---------|---------|-----------|---------|-------------|
> | [B]-2023| Stage 1 | HLR       | -10.02  | -          |
> |         |         | Ours      | **-12.01** | 2.01%   |
> |         | Stage 2 | +Adaptor1 | -14.89  | -          |
> |         |         | +Ours     | **-16.95** | 2.06%   |
> | [C]-2023 | Stage 1 | BLR       | -4.74   | -          |
> |         |         | Ours      | **-12.01** | 7.27%   |
> |         | Stage 2 | +Adaptor2 | -10.94  | -          |
> |         |         | +Ours     | **-15.26** | 4.32%   |
> | [D]-2023 | Stage 1 | HLR       | -9.66   | -          |
> |         |         | Ours      | **-12.01** | 2.35%   |
> |         | Stage 2 | +Adaptor3 | -13.57  | -          |
> |         |         | +Ours     | **-16.06** | 2.49%   |
>
> [A]-2023: Downscaled representation matters: Improving image rescaling with collaborative downscaled images, ICCV 2023.
>
> [B]-2023: Universal deep image compression via content-adaptive optimization with adapters, WACV, 2023
>
> [C]-2023: Dec-adapter: Exploring efficient decoder-side adapter for bridging screen content and natural image compression, ICCV, 2023
>
> [D]-2023: Dynamic low-rank instance adaptation for universal neural image compression, ACMMM, 2023

---

> ### Author Response · Authors · 2024-11-19
> **Thank you; Address your concerns [2/2]**
>
> **W2:** *It would be better to report the image compression results on top of popular image reconstruction benchmarks, such as Set5, Set14, BSD100, Urban100, etc.*
>
> **Response:** Thanks for your comments. We highly appreciate your advice and report the results on your suggested benchmark datasets. As illustrated in the following table, our proposed method outperforms baseline methods by an obvious margin, which reflects a good generalization ability of our proposed method on wide datasets. **Please refer to the orange texts in Page 16 of the revised manuscript.**
>
> *Table 6. Comparison of our method with existing latent refinement approaches in terms of BD-rate savings (% ↓) using base models without test-time adaptation. Smaller values indicate better performance.*
>
> | Method    | Set5    | Set14   | BSD100  | Urban100 | Average ↓ |
> |-----------|---------|---------|---------|----------|-----------|
> | AR-CM     | 0       | 0       | 0       | 0        | 0         |
> | + BLR     | -13.27  | -15.32  | -5.92   | -16.25   | -12.69    |
> | + HLR     | -15.01  | -18.29  | -6.61   | -21.42   | -15.33    |
> | + Ours    | **-21.47** | **-24.88** | **-11.23** | **-26.68** | **-21.06** |
> ***
> ***
> **W3:** *This paper is hard to follow and the writing could be further improved. It would be better to explicitly highlight the key contributions of this paper.*
>
> **Response:** Thank you for your comment.
>
> First, in order to improve the readability, we have added a preliminary section to explain the image compression process before the theoretical analysis of existing methods. We hope this background knowledge can enhance reader's understanding of the overall paper. **Please refer to the blue texts in Page 3 of the revised manuscript.**
>
> Second, by following your suggestion, we have added the summarization of our contributions in the introduction section. **Please refer to the orange texts in Page 3 of the revised manuscript.**
> ***
> ***
> **W4:** *The compared methods BLR and HLR are too old since both are published before 2021. It would be better to compare with more recent work, such as [A]. [A] Downscaled representation matters: Improving image rescaling with collaborative downscaled images, ICCV 2023.*
>
> **Response:** Thanks for your comment!
>
> First, it should be noted that, in the image compression community, these two latent refinement approaches play a fundamental role in recent state-of-the-art TTA-IC models (such as [B]-2023,[C]-2023, and [D]-2023), especially the HLR. Compared with the emergence of various decoder refinement variants, the latent refinement is barely tailored to cross-domain scenarios, exhibiting underexplored space. This is one of our motivations in this paper. To the best of our knowledge, BLR and HLR are only two kinds of latent refinement baselines in the image compression community.
>
> Second, at a high level, the baseline you suggested, namely HCD, is related to the TTA-IC task, since both HCD and our proposed method directly optimize the mediate representations (*i.e.,* downscaled image representation for HCD and latent representations of compressed image for our method) without model modification at test time. However, there are several significant differences between image rescaling and image compression, impeding the application of image rescaling methods to image compression problems, as follows:
>
> - **Low-resolution (LR) image vs. Compressed bitstream:** HCD mainly focuses on image rescaling problems that downscale a high-resolution image to a good low-resolution image, which remains visible to humans. In contrast, image compression usually aims to transform the image, regardless of whether it is high-resolution or low-resolution, into a bitstream (invisible to humans) that can be transmitted between devices. These bitstreams are obtained by quantization and entropy coding steps (refer to Figure 2) for latent representations. Different outputs pose a difference between image rescaling and image compression.
>
> - **Image rescaling does not involve quantification and entropy coding:** HCD directly learns the optimal perturbations to improve the downscaled image representation. Unlike image compression, HDC has no entropy coding step that compresses the latent representation to the bitstream. Typically, advanced image compression models utilize hyperprior models to approximate the joint probability of latent representation and side information for entropy coding. Since image rescaling methods such as HCD do not refer to such probability models for entropy coding, the direct application of HCD to image compression is infeasible.
>
> Last but not least, in light of its valuable contributions (the first attempt to optimize the downscaled representation instead of the model), we will cite this paper to provide detailed discussions in the related work of the revised paper. **Please refer to the orange text in Page 3 of the revised manuscript.**

---

> ### Author Response · Authors · 2024-11-25
> **Thank you for your comments**
>
> Dear Reviewer ShHq,
>
> Thank you for taking the time and effort to review our paper again. We have carefully addressed your valuable comments, such as  "The experimental results of this plug-and-play method are relatively weak ...... " .
>
> As the public discussion phase is coming to an end soon, we kindly invite any further comments or suggestions you may have. We sincerely appreciate your efforts, which have significantly contributed to improving our manuscript.
>
> Best regards,
>
> The Authors

---

> > ### Comment · Reviewer_ShHq · 2024-11-28
> > **Thanks for the response**
> >
> > Thanks for the response.
> >
> > After reading the response and the comments from other reviewers, I find that my major concern on ``insufficient comparisons'' is still not well addressed. Interestingly, it seems a common concern that is also raised by Reviewer GYDX. The authors pay too much attention to discussing the differences between the image compression (IC) task and the others. Nevertheless, what should be more important is the technical contributions in test-time adaptation. In fact, a lot of good test-time adaptation works have been used in diverse tasks and most of them have released their code for reproduction. In my opinion, most test-time adaptation methods can be applied to the IC task and the comparisons with these methods would be interesting. If not, I am happy to see the reasons.
> >
> > Best,
> >
> > Reviewer ShHq

---

> ### Author Response · Authors · 2024-11-29
> **Thanks for your feedback**
>
> Dear Reviewer ShHq,
>
> Thanks for reading our responses and providing follow-up questions. Generally, most conventional TTA approaches cannot be directly applied to IC tasks. We provide detailed justifications as follows:
> ***
> Existing mainstream TTA methods, including data augmentation/perturbation-based approaches and pseudo-labeling-based approaches, typically cater to classification or regression tasks. However, to achieve a good rate-distortion (R-D) performance on IC tasks, one needs to uniquely consider both
> - *The rate cost of intermediate features after quantization and entropy coding.*
> - *The reconstruction quality from the compressed bitstream.*
>
> The fundamental difference in objectives between common TTA tasks and TTA-IC tasks raises unique challenges and specific learning behaviors for the design of TTA-IC algorithms, and **conventional TTA approaches cannot be directly applied to IC tasks**.
>
> More specifically, data augmentation/perturbation employs various data transformations (e.g., noise) to generate new samples/features, which increases the variance (entropy) of latent representations. This conflicts with IC tasks that minimize the entropy to acquire the most compact latent representations. Besides, pseudo-labeling-based approaches roughly rely on either weak-and-strong augmentation-based frameworks or data class prototypes, making them unadaptable for IC tasks due to the inherent entropy increase problem and unavailable compression-related prototypes. Lastly, we have conducted additional experiments comparing the common TTA approach (MEMO [1], RMT [2]), TTA for rescaling(HCD [3]), and TTA-IC (our method). The results on summarized below:
>
> Table1: The evaluation is measured in terms of average BD-rate savings (% ↓)
> | Method| Kodak | SIQAD   | SCID | CCT| Pixel   | DomainNet | Average ↓ |
> |---|----|---|---|---|----|----|---|
> | **AR-CM**                        | 0                 | 0       | 0       | 0       | 0       | 0         | 0         |
> | + MEMO                           | 1.98              | -7.12   | -5.34   | -5.50   | -5.05   | -7.40     | -5.23     |
> | + RMT                            | -7.50             | -8.12   | -8.25   | -6.83   | -7.92   | -9.28     | -7.98     |
> | + HCD                            | -8.01             | -8.65   | -8.92   | -7.21   | -8.78   | -10.37    | -8.59     |
> | + BLR                            | -6.75             | -18.75  | -18.31  | -25.33  | -26.22  | -18.78    | -21.41    |
> | + HLR                            | -12.73            | -21.28  | -15.11  | -23.08  | -31.76  | -15.61    | -21.36    |
> | + **Ours**                       | **-16.11**        | **-24.71** | **-22.63** | **-28.26** | **-35.89** | **-23.52** | **-27.00** |
> | **Hyperprior**                   | 0                 | 0       | 0       | 0       | 0       | 0         | 0         |
> | + MEMO                           | 1.12              | -5.05   | -3.35   | -3.45   | -3.23   | -5.30     | -3.25     |
> | + RMT                            | -5.75             | -6.10   | -6.25   | -4.45   | -5.98   | -7.57     | -6.02     |
> | + HCD| -6.51 | -4.78   | -7.12   | -7.35   | -6.89   | -8.99     | -6.95     |
> | + BLR | -6.59             | -13.16  | -16.71  | -18.91  | -15.12  | -22.99    | -15.41    |
> | + HLR  | -11.32            | -13.94  | -14.59  | -13.82  | -18.89  | -16.68    | -14.87    |
> | + **Ours** | **-14.82**        | **-20.95** | **-23.51** | **-23.58** | **-22.22** | **-23.32** | **-20.78** |
>
> Table2: Bit-per-pixel (BPP) consumption and PSNR on SIQAD dataset with AR-CM model (quality = 0)
> |   | MEMO |RMT| HCD | BLR | HLR | Ours |
> |---|---|---|----|----|----|---|
> | BPP↓| 0.591| 0.576| 0.599 | 0.316  | 0.386 | 0.320|
> | PSNR | 26.41 | 26.87 | 27.01| 26.06 | 26.98 | 26.98 |
>
> As shown in Table1, these conventional TTA approaches cannot achieve satisfactory results on IC tasks. Moreover, from Table2 we can see that while common TTA methods behave competitively in PSNR (even better for HCD), they compromise bit-per-pixel consumption. This verifies our explanation that data augmentation/perturbation in image or feature domain would increase the entropy consumption of latent representations, enhancing the difficulty of R-D optimization.
>
> **We will add these comparisons and discussions in Appendix. Thanks for your valuable insights!**
> ***
> **Reference**
>
> [1] Test Time Robustness via Adaptation and Augmentation. NeurIPS, 2022. We calculate the energy score of multiple reconstructed results (\#augmented views: 8), as an alternative to classification entropy.
>
> [2] Robust Mean Teacher for Continual and Gradual Test-Time Adaptation. CVPR, 2023. We calculate the MSE between two reconstructed results as a consistency regularization.
>
> [3] Downscaled representation matters: Improving image rescaling with collaborative downscaled images, ICCV 2023. To adapt the HCD to IC tasks, the LR, and HR images are regarded as the latent representation and compressed image, respectively.

---

> > ### Author Response · Authors · 2024-12-02
> > **Thank you**
> >
> > Dear Reviewer ShHq ,
> >
> > As the discussion period is ending soon, we would like to send a kind reminder about our latest responses.
> >
> > We have addressed your concerns in detail and will incorporate your suggestions into our revisions. If there are still remaining concerns, we will do our best to provide clarifications as soon as possible. Otherwise, we look forward to your positive feedback.
> >
> > Once again, we appreciate your time and consideration.

---

> > > ### Comment · Reviewer_ShHq · 2024-12-03
> > > **Additional results and comparisons make sense to me**
> > >
> > > Thanks for providing additional comparisons. These results make sense to me. I would raise my rating to 6 and highly recommend the authors to include these results into the paper.

---

> > > > ### Author Response · Authors · 2024-12-03
> > > > **Thank you**
> > > >
> > > > Thanks for your positive feedbacks and constructive suggestions. We will integrate these results into final manuscript.
> > > >
> > > > Thanks for your time again.

---

### Official Review · Reviewer_7uht · 2024-11-02

**Soundness:** 3
**Presentation:** 3
**Contribution:** 2
**Rating:** 6
**Confidence:** 5

**Summary:**

Current Test-Time Adaptation Image Compression (TTA-IC) methods improve rate-distortion (R-D) performance on cross-domain tasks by refining both latent variables and decoders in a two-step process. However, latent refinement has not been specifically optimized for cross-domain scenarios. The authors conduct theoretical analyses to identify a mismatch between refined Gaussian conditionals and hyperprior distributions in the vanilla Hybrid Latent Refinement (HLR), leading to poorer joint probability approximations and higher rate consumption. To address this issue, they introduce a Bayesian approximation-based distribution regularization technique that enhances joint probability modeling in a plug-and-play manner. Extensive experiments across six in-domain and cross-domain datasets demonstrate that the proposed method outperforms existing latent refinement approaches in R-D performance.

**Strengths:**

1. The paper effectively identifies the limitations of existing Hybrid Latent Refinement (HLR) methods in cross-domain image compression, particularly highlighting the mismatch between refined Gaussian conditionals and hyperprior distributions.

2. It provides a thorough theoretical analysis using marginalization approximation, establishing a solid foundation for the proposed improvements.

3. The introduction of a Bayesian approximation-based distribution regularization technique successfully addresses the identified mismatch, enhancing joint probability approximation.

4. The method shows improvements in rate-distortion (R-D) performance compared to other latent refinement approaches, validating its effectiveness.

**Weaknesses:**

1. The paper does not include a comparison of GPU memory usage and GFLOPs, which are crucial metrics for evaluating the complexity of various methods. Including these metrics would illustrate the trade-off between performance gains and computational complexity more clearly. It would be beneficial to present memory usage across different datasets, especially since DVI is introduced during optimization, making it reasonable to assess the additional complexity incurred.

2. In [1], the posterior distribution of $\hat{z}$ is assumed to be a uniform distribution. It is unclear whether directly inferring $\hat{z}$ through a hyper analysis transform in out-of-distribution (OOD) scenarios is the better choice? Because this approach not only avoids incurring a larger $\Delta H$ but also naturally matches its posterior distribution during pre-training.

3. The paper introduces a regularization term based on HLR in TTA-IC. While this contribution is valuable, its impact on the community appears moderate.

[1] Ballé, J., Minnen, D., Singh, S., Hwang, S. J., & Johnston, N. (2018). Variational image compression with a scale hyperprior. arXiv preprint arXiv:1802.01436.

**Questions:**

Will the authors make their code public to assist the community upon acceptance?

---

> ### Author Response · Authors · 2024-11-19
> **Thank you; Address your concerns [1/2]**
>
> Dear Reviewer,
>
> Thank you for dedicating your time and effort to reviewing our paper. We are glad that you find our approach well-motivated, providing a thorough theoretical analysis and exhibiting promising performance. We carefully considered your suggestions and conducted additional experiments to improve our paper. Please find our point-to-point responses below and corresponding modifications in the revised manuscript **(marked by the green)**.
> ***
>
> **W1:** *The paper does not include a comparison of GPU memory usage and GFLOPs.*
>
> **Response:** Thanks for your constructive comment.
>
> Since the GPU memory usage and GFLOPs rely on the resolution of input images, we choose two typical datasets with common image resolution, including Kodak (512$\times$768) and CCT (1080$\times$1920), to evaluate the computational complexity. As illustrated in the following results,   the introduction of DVI indeed slightly incurs additional computational complexity compared with baseline methods.
>
> It also should be noted that such a latent refinement stage is usually conducted on the encoding side of the server, which means that there is no effect on the real-time decoding on edge devices, making our proposed method applicable to real-world scenarios. Moreover, our proposed method has no model modification to obtain probability representation based on DVI. This simplification also enhances its usability.
>
> **Please refer to the blue text in Page 17 of the revised manuscript.**
>
> *Table 7. Average adaptation cost of different methods on Kodak and CCT datasets.*
> | Metric | BLR | HLR | Ours |
> |--------|-----|-----|------|
> | **GPU Memory (GB)** | 0.72/3.67 | 0.74/3.92 | 0.79/3.98 |
> | **GFLOPs** | 98.98/525.81 | 101.23/563.24 | 110.58/587.47 |
> ***
> ***
>
> **W2:** *In [1], the posterior distribution of is assumed to $\hat{z}$ be a uniform distribution. It is unclear whether directly inferring
>  through a hyper analysis transform in out-of-distribution (OOD) scenarios is the better choice?*
>
> **Response:** Thank you for your interesting comment.
>
> Although it is feasible to conduct a direct inferring through a hyper analysis transform in out-of-distribution (OOD) scenarios, the result is significantly behind the proposed distribution regularization. The result (-4.84) seems to be similar to the setting with DL=3 and $\beta=10$ (-4.95) and the deterministic distribution regularization (-3.54).
>
> We conjecture that the direct inferring may be a deterministic over-regularization method. Since the optimization of latent representation is directly affected by the reconstruction loss and rate cost of latent representation, the usage of direct inferring result $z$ still cannot be ensured to lead to minimal rate cost of side information and appropriate conditional distribution $p(y|z)$. Moreover, the optimization of side information $z$ uses the direct inferring result may result in a potential optimization difficulty, due to continually variational $z$ obtained by a hyper analysis transform.
> Therefore, it seems that an appropriate regularization like the proposed distribution regularization is necessary to alleviate the optimization difficulty between latent representation and side information, leading to more matched joint probability.
>
> *Table 4. Ablation study of proposed distribution regularization L_DR (BD-rate % ↓) on SIQAD. DL denotes number of dropout layers. †: assume σ=1 without Bayesian approximation. ‡: minimizing Eq. BPP.*
>
> | Method | β=0.1 | β=1 | β=10 |
> |--------|-------|-----|------|
> | **Baseline (HLR)** | 0 (anchor) | - | - |
> | **+ L_DR (DL=1)** | -10.01 | -10.02 | -7.07 |
> | **+ L_DR (DL=2)** | -10.03 | -9.62 | -5.25 |
> | **+ L_DR (DL=3)** | **-10.06** | -9.61 | -4.95 |
> | **+ Deterministic L_DR†** | -3.54 | - | - |
> | **+ Direct Inferring** | -4.84 | - | - |
> | **+ BBC‡** | 0.15 | - | - |
>
> **Thanks for your interesting comment! We have added this dicussion. Please refer to the green text in Page 17 of the revised manuscript.**
> ***
> ***
>
> **W3:** *The paper introduces a regularization term based on HLR in TTA-IC. While this contribution is valuable, its impact on the community appears moderate.*
>
> **Response:** Thank you for your comment.
>
> While the introduction of a regularization term based on HLR in TTA-IC may seem moderate, its implications are significant within the scope of ICLR. Our method addresses a critical gap in cross-domain image compression, enhancing the rate-distortion performance, which is crucial for real-world applications like streaming and storage. By providing a theoretical analysis foundation and demonstrating practical improvements across diverse datasets, our approach not only advances the state-of-the-art but also offers a versatile solution that can be integrated into existing TTA-IC systems. This flexibility and practical impact underscore the broader relevance and potential of our work to drive future research and applications in the field.

---

> > ### Comment · Reviewer_7uht · 2024-11-25
> > **Thanks for the response**
> >
> > Thanks for the authors' response. I will keep my score.

---

> > > ### Author Response · Authors · 2024-11-25
> > > **Thank you**
> > >
> > > Dear reviewer,
> > >
> > > Thank you for dedicating your time and effort to reviewing our paper again. Your positive comments and constructive suggestions intensively improve our paper.
> > >
> > > Thanks!
> > >
> > > Best wishes,
> > >
> > > The authors

---

> ### Author Response · Authors · 2024-11-19
> **Thank you; Address your concerns [2/2]**
>
> **W4:** Will the authors make their code public to assist the community upon acceptance?
>
> **Response:** Yes, we are very glad to share the code upon acceptance. **In the revised manuscript, we have added a placeholder to the abstract to clarify the availability of our code. Please refer to the green text in Page 1 of the revised manuscript.**

---

### Official Review · Reviewer_GYDX · 2024-11-02

**Soundness:** 3
**Presentation:** 3
**Contribution:** 3
**Rating:** 6
**Confidence:** 4

**Summary:**

The authors propose a new latent refinement method to enhance the rate-distortion (R-D) performance for test-time adaptation image compression. They theoretically analyze that existing methods require higher rate consumption to estimate the marginal probability of the latent variables in cross-domain scenarios. To address this, the authors incorporate a new distribution regularization term into the R-D objective, promoting a better approximation of the latent representation. Experimental results across six cross-domain datasets demonstrate the effectiveness of the proposed approach, showing its superiority over existing methods. Please see my detailed comments below.

**Strengths:**

1. The authors offer a theoretical analysis highlighting limitations of existing methods, which informs their introduction of a distribution regularization technique. This technique effectively adapts the compression model to test data with domain shifts.
2. They provide detailed analysis results on the entropy curves of different probabilities about different methods. The results in Figure 4 demonstrate that the proposed method can lead to a good convergence of rate consumption and better R-D performance.
3. Experimental results demonstrate that the proposed method can be easily integrated with existing methods to enhance R-D performance across cross-domain image datasets.

**Weaknesses:**

1. The test datasets used in the experiments contain a limited number of images, which may affect the robustness of the results. To strengthen the findings, please consider providing additional results on larger-scale datasets.
2. It would be helpful to include a preliminary section explaining the image compression process before the theoretical analysis of existing methods. This section could clarify the steps involved in obtaining the latent variable y and hyper-latent variable z (side information), enhancing readers' understanding of the compression model and the subsequent theoretical analysis.
3. In addition to the adaptation cost of different methods, it is better to provide the compression cost of the proposed methods, which would offer a more comprehensive view of its performance.
4. The authors do not provide a limitation analysis of the proposed methods. For instance, the long adaptation time may limit the proposed methods to be used in real-time applications.
5. More discussion on recent test-time adaptation works would be beneficial to place this study in context. Relevant works to consider include, but are not limited to, [A-E].

[A] Test-Time Training Can Close the Natural Distribution Shift Performance Gap in Deep Learning Based Compressed Sensing, ICML 2022.

[B] Efficient Test-Time Model Adaptation without Forgetting, ICML 2022.

[C] Tent: Fully Test-Time Adaptation by Entropy Minimization, ICLR 2021.

[D] Efficient Test-Time Adaptation for Super-Resolution with Second-Order Degradation and Reconstruction, NeurIPS 2023.

[E] Test Time Adaptation for Blind Image Quality Assessment, ICCV 2023.

**Questions:**

1. In Table 3, the results indicate that increasing the number of dropout layers can negatively impact image compression performance. Please add further discussion on how to effectively approximate the distribution regularization to avoid such performance degradation.

---

> ### Author Response · Authors · 2024-11-19
> **Thank you; Address your concerns [1/2]**
>
> Dear Reviewer,
>
> Thank you for dedicating your time and effort to reviewing our paper. We sincerely appreciate your positive remarks regarding our theoretical analysis, effectiveness for domain shifts, detailed analysis using entropy curves, and plug-and-play advantage.    We carefully considered your suggestions to improve our paper. Please find our point-to-point responses below and corresponding modifications in the revised manuscript **(marked by the blue (except the first response))**.
> ***
> **W1:** *The test datasets used in the experiments contain a limited number of images, which may affect the robustness of the results. To strengthen the findings, please consider providing additional results on larger-scale datasets.*
>
> **Response:** Thanks for your comments.
>
> We have adopted your suggestion to conduct additional experiments on larger-scale datasets. Specifically, we utilize BSD100 and Urban100 benchmark datasets for evaluation, where each dataset has 100 high-resolution images. As illustrated in the following table, our proposed method can surpass baseline methods in terms of the BD-rate. Such performance gains show a good generalization ability of our proposed method on diverse datasets, exhibiting its usability in the real world. **Please see the orange text in Page 16 of the revised manuscript.**
>
> *Table 6. Comparison of our method with existing latent refinement approaches in tems of BD-rate savings (% ↓) using base models without test-time adaptation. Smaller values indicate superior performance.*
> | Method | BSD100 | Urban100 | Average ↓ | Improvement |
> |--------|---------|-----------|-----------|-------------|
> | AR-CM  | 0       | 0         | 0         | -           |
> | + BLR  | -5.92   | -16.25    | -11.08    | 11.08%      |
> | + HLR  | -6.61   | -21.42    | -14.01    | 14.01%      |
> | + Ours | **-11.23** | **-26.68** | **-18.95** | **18.95%** |
> ***
> ***
> **W2:** *It would be helpful to include a preliminary section explaining the image compression process before the theoretical analysis of existing methods. This section could clarify the steps involved in obtaining the latent variable y and hyper-latent variable z (side information), enhancing readers' understanding of the compression model and the subsequent theoretical analysis.*
>
> **Response:** Thanks for your suggestion! It is indeed useful to include a preliminary section to explain the image compression process in more detail. This can enhance the readability of this paper to wider audiences. To this end, we have added the preliminary section in the revised manuscript. **Please refer to the blue texts in Page 3 of the revised manuscript.**
> ***
> ***
> **W3:** *In addition to the adaptation cost of different methods, it is better to provide the compression cost of the proposed methods, which would offer a more comprehensive view of its performance.*
>
> **Response:** Thanks for your comment. We have adopted your suggestion to compute the adaptation cost of different methods in terms of GPU memory usage and GFLOPs. As illustrated in the following table, our proposed method has a minor increase in adaptation cost, which is reasonable as the dropout variational inference is introduced during optimization for better R-D performance.
>
> Notably, although our proposed method increases the adaptation cost, such a latent refinement stage is usually conducted on the encoding side of the server, which means that there is no effect on the real-time decoding on edge devices, making our proposed method applicable to real-world scenarios. Moreover, our proposed method has no model modification to obtain probability representation based on dropout variational inference. This simplification also enhances its usability.
>
> **Please refer to the blue texts in Page 16 of the revised manuscript.**
>
> *Table 7. Average adaptation cost of different methods on Kodak and CCT datasets.*
> | Metric | BLR | HLR | Ours |
> |--------|-----|-----|------|
> | **GPU Memory (GB)** | 0.72/3.67 | 0.74/3.92 | 0.79/3.98 |
> | **GFLOPs** | 98.98/525.81 | 101.23/563.24 | 110.58/587.47 |
> ***
> ***
> **W4:** *The authors do not provide a limitation analysis of the proposed methods. For instance, the long adaptation time may limit the proposed methods to be used in real-time applications.*
>
> **Response:** Thanks for your comment. We have adopted your advice to add a limitation discussion of our proposed method in the revised manuscript, as follows.
>
> *Due to the introduction of the dropout variational inference, our proposed method has a minor increase in adaptation cost compared with baseline methods. Although such a latent refinement stage is usually conducted on the server without effects on the real-time decoding on edge devices, developing a more efficient solution and using a more powerful GPU on the server can further reduce the adaptation time in the future.*
>
> **Please refer to the blue texts in Page 11 of the revised manuscript.**

---

> ### Author Response · Authors · 2024-11-19
> **Thank you; Address your concerns [2/2]**
>
> **W5:** *More discussion on recent test-time adaptation works would be beneficial to place this study in context. Relevant works to consider include, but are not limited to, [A-E].*
>
> **Response:** Thanks for your comment. We have adopted your advice to add a discussion for these valuable works you suggested, in the related work section. We highly appreciate your constructive comments that intensively enhance the readability of our paper! **Please refer to the blue text in Page 3 of the revised manuscript.**
> ***
> ***
> **W6:** *In Table 3, the results indicate that increasing the number of dropout layers can negatively impact image compression performance. Please add further discussion on how to effectively approximate the distribution regularization to avoid such performance degradation.*
>
> **Response:** Thanks for your comment. We have adopted your suggestion to conduct a deeper discussion. Specifically, as illustrated in Table 4, the best performance is achieved when the number of the dropout layer is 3. To avoid performance degradation, it is better to construct more dropout layers with more accurate probability representation and set a lower regularization coefficient (such as 0.1) to alleviate the over-regularized downside. **Please refer to the blue text in Page 11 of the revised manuscript.**

---

> ### Author Response · Authors · 2024-11-26
> **Thank you**
>
> Dear Reviewer GYDX,
>
> Thank you for taking the time and effort to review our paper again. We have carefully addressed your valuable comments.
>
> As the public discussion phase is coming to an end soon, we kindly invite any further comments or suggestions you may have. We sincerely appreciate your efforts, which have significantly contributed to improving our manuscript.
>
> Best regards,
>
> The Authors

---

> > ### Comment · Reviewer_GYDX · 2024-12-01
> > **Thanks for the response**
> >
> > Thanks for the authors' thoughtful responses. The additional results and discussions seem fair. I will keep my original score.

---

> ### Author Response · Authors · 2024-12-01
> **Thanks for your feedback**
>
> Dear Reviewer GYDX,
>
> Thanks for your positive feedbacks and constructive suggestions, which have intensively improved our paper.
>
> Thanks for your time again.
>
> Best wishes,
>
> The authors

---

### Official Review · Reviewer_5FRR · 2024-11-03

**Soundness:** 2
**Presentation:** 2
**Contribution:** 2
**Rating:** 6
**Confidence:** 3

**Summary:**

This paper addresses a latent refinement techqniue in test-time adaptation methods for image compression.
Especially, the paper focuses on methods that use a hyper-prior model, where the image is encoded into a latent variable $y$ and a hyper-latent variable $z$ in a hierarchical way.
The paper analyzes the issue of recent latent refinement methods not generalizing well in cross-domain settings, arguing the cause to a mismatch between the hyper-prior $p(z)$ and the gaussian conditional $p(y|z)$ used in refinement methods.
Based on this conjecture, the paper proposes a distribution regularization term that uses dropout variational inference, claiming it can improve probability approximation.
Experimental results show that the proposed method continuously improves latent refinement and demonstrates effectiveness even when integrated into state-of-the-art methods that jointly utilize latent and decoder refinement.

**Strengths:**

1. The proposed method can be applied without altering any model parameters, making optimization easy and immune to catastrophic forgetting.
2. The paper tries providing detailed discussions on the proposed method, both in a principled and empirical way.
3. The experimental results consistently demonstrate the effectiveness of the proposed method in cross-domain image compression across different settings: (1) with only latent refinement, (2) integrated with state-of-the-art TTA-IC methods, which further include decoder adaptation, and (3) on medical images.

**Weaknesses:**

Most importantly, as i understand, the theoretical analysis and subsequent derivation of the proposed method, which are claimed as major contribution in this paper, contain many unclear and misleading aspects.
1. (L173, 193) The definition of "optimal probability representation" is not clearly defined, throughout the paper.
2. The entropy function, i.e., $H(X)=-\mathbb{E}_{x\sim p(x)}[\log p(x)]$, is interpreted in the form of log probability, and it seems that the expectation term is not considered in the analysis and derivation.
3. In the Proposition 1, assuming an optimal joint probability approximation of the true marginal distribution $p(y^*)=\int p(y,z)dz$, the proposition simply interprets $H(Y,Z) - H(Y^*) = H(Z|Y) + H(Y) - H(Y^*) = H(Z|Y)$ when $H(Y) = H(Y^*)$. However, the assumption $p(y^*)=\int p(y,z)dz$ may not induce equations (7,8), which can be misleading.
4. In the proof of Proposition 2, the authors argue that the entropy bottleneck $p(z_t)$ and the posterior $p(y_t|z_t)$ are learned from the source dataset (L212, source image correlated) and may not work effectively. While this makes sense, it is not a theoretical analysis, in my point of view.
5. For Corollary 1, it is unclear whether $z^*$ and $y^*$ in equation 13 correspond to in-domain or cross-domain cases. In the cross-domain case with $y^*_t$ and $z_t^*$ as in Proposition 2, equation 13 is as follows: $\triangle H = H(Y, Z) - H(Y^*_t) > H(Z^*|Y^*)$, which holds according to equations (11, 12) only if $H(Y^*_t)=H(Y^*)$ and $H(Z^*_t|Y^*_t)=H(Z^*|Y^*)$. However, there is no guarantee for the condition.
6. Importantly, in Section 3.3, the paper proposes distribution regularization using equation 13, interpreting the first three terms in equation 17 as corresponding to $H(Y|Z), H(Z), H(Z^*|Y^*)$ (the three terms at the front) in equation 13. However, it is unclear how $H(Z^*|Y^*)=\mathbb{E}_{y^*,z^*\sim p(y^*,z^*)}[p(z^*|y^*)]$ leads to $-\log p(z^m_t|y^m_t)$ in the optimization objective. Note that $H(Z^*|Y^*)$ includes expectation from $p(y^*,z^*)$.

In conclusion, the theoretical analysis in this paper seems to have many over claimed or misclaimed elements.

**Questions:**

* Please address the weakness above.
* The writing is a bit difficult to understand, especially in the discussion parts of both the methodology and experiment sections

---

> ### Author Response · Authors · 2024-11-16
> **Thank you; Address your concerns [1/3]**
>
> Dear Reviewer,
>
> Thank you for dedicating your time and effort to reviewing our paper. We are glad that you think our method can make optimization easy, have detailed discussion, and can achieve good results in multiple settings. We took your suggestions very seriously, especially addressing your confusion about our theoretical analysis. Please see below for our point-to-point responses and corresponding modifications in the revised manuscript **(marked by the red)**.
>
> **Note: If the equation is displayed by original markdown codes, please refresh this page to reactivate the compiler.**
>
> ***
> ***
> **W1:** The definition of "optimal probability representation" is not clearly defined, throughout the paper.
>
> **Response:** We agree that the definition of "optimal probability representation" is unclear. In a nutshell, the optimal probability representation of learned encoding distributions, *e.g.,* $p(y|z,\theta_{h_{e}})$ and $p(z|\varphi)$, converge to their true underlying distributions with minimum rate cost according to Shannon's entropy theorem. A formal definition is represented in the revised manuscript as follows.
> ***
> **Definition 1** (_Optimal Probability Representation_). By minimizing Eq. 3 (R-D objective), there exist learnable parameters $\theta^*_{h_{e}}$ and $\varphi^*$ that achieve minimum rate cost for $p(y|z)$ and $p(z)$. According to Shannon's entropy theorem, the learned encoding distributions $p(y|z)$ and $p(z)$ thus converge to optimal probability representations $p(y^*|z^*)$ and $p(z^*)$ that exactly match the true underlying distributions:
>
> $$
> p(y \mid z) = p(y \mid z, \theta_{h_e}^*) =p(y^* \mid z^*) \quad p(z) = p(z \mid \varphi^*) = p(z^*).
> $$
> ***
> This explicit definition of optimal probability representation involves Proposition 1, Proposition 2, and Corollary 1. It is desirable to eliminate your confusion upon these derivations concurrently.
> ***
> ***
> **W2:** The entropy function, i.e., $H(x)=-\mathbb{E}_{x\sim p(x)}[\log p(x)]$, is interpreted in the form of log probability, and it seems that the expectation term is not considered in the analysis and derivation.
>
> **Response:** Thanks for your comments. For paper readability and notation simplicity, we omit the expectation term in the entropy calculation, though it is implicitly considered in our derivation. To eliminate such confusion, we reformulate the expectation term of the entropy function in the revised manuscript.
> ***

---

> ### Author Response · Authors · 2024-11-16
> **Thank you; Address your concerns [2/3]**
>
> ***
> ***
> **W3:** In the Proposition 1, assuming an optimal joint probability approximation of the true marginal distribution $p(y^*)=\int p(y,z)dz$, the proposition simply interprets $H(Y,Z) - H(Y^*)=H(Z|Y)+H(Y)-H(Y^*)=H(Z|Y)$ when $H(Y)=H(Y^*)$. However, the assumption $p(y^*)=\int p(y,z)dz$ may not induce equation (7,8), which can be misleading.
>
> **Response:** Thanks for your comment!
>
> First, while the decomposition of the joint distribution you suggested is feasible, leading to $H(y,z)=H(z|y)+H(y)$ related to $p(z|y)$ and $p(y)$, such decomposition may not directly align the R-D objective in Eq.  (3), *i.e.*, $p(y|z)$ and $p(z)$ that are more meaningful in the neural compression community due to their direct correlations with rate cost. More importantly, $p(y|z)$ and $p(z)$ help us analyze the rate cost change of learned encoding distributions $p(y|z,\theta_{h_{e}})$ and $p(z|\varphi)$ when $\theta_{h_{e}}$ and $\varphi$ are fixed in latent refinement scenarios and domain shifts are incurred. Therefore, when $H(y,z)=H(y|z)+H(z)$, Eqs. (7) and (8) can be induced.
>
> Second, Lemma 1 is recalled to clarify the joint probability $p(y,z) = p(y|z) \cdot p(z)$ rather than $p(y^*)=\int p(y,z)dz$. Due to the condition of optimal joint probability approximation of true marginal distribution (means optimal probability representation as in Definition 1), $p(y|z)=p(y^*|z^*)$ and $p(z) = p(z^*)$ make Eqs. (7) and (8) hold. Therefore, Eqs. (7) and (8) can be induced.
>
> The unclear decomposed motivation and the recalling of Lemma 1 may impede your understanding, A revised version of Proposition 1 is represented in the revised manuscript as follows. We hope your confusion can be eliminated.
> ***
> **Proposition 1**
> Let $y$ and $z$ be the latent and hyper latent variables, and these variables with the asterisk be their optimal representations. In the context of in-domain image compression, if an optimal joint probability approximation of true marginal distribution can be achieved by minimizing Eq. (3), the extra rate consumption of marginalization approximation is
> $$
> \Delta H^* = H(y,z) - H(y^*) = \mathbb{E}_{y,z\sim p(y,z)}[- \log p(z|y)]  \tag{4}
> $$
>
> **Proof**
> On the one hand, with joint probability and the Bayesian rule $p(y^*)=\frac{p(y^*|z^*)p(z^*)}{p(z^*|y^*)}$, we have
>
> $$
> H(y,z) = \mathbb{E}_p[- \log p(y|z)-\log p(z)],  H(y^*)  = \mathbb{E}_p[- \log p(y^*|z^*) - \log p(z^*) + \log p(z^*|y^*)]. \tag{5}
> $$
>
> Then, we have
> $$
> \Delta H^*=\mathbb{E}_p[-\log p(y|z) - (-\log p(y^*|z^*))] + \mathbb{E}_p[-\log p(z)- (-\log p(z^*))] - \log p(z^*|y^*). \tag{6}
> $$
>
> On the other hand, as there exist optimal probability representations for $p(y|z)$ and $p(z)$ for in-domain image compression by minimizing Eq. (3), we have
>
> $$
> \mathbb{E}_p[-\log p(y|z) - (-\log p(y^*|z^*))] = 0, \quad s.t. \quad p(y|z)=p(y^*|z^*), \tag{7}
> $$
> $$
> \mathbb{E}_p[-\log p(z)- (-\log p(z^*))] = 0,\quad s.t. \quad p(z) = p(z^*), \tag{8}
> $$
>
> Thus,
> $$
> \Delta H^* = \mathbb{E}_p[- \log p(z^*|y^*)] = \mathbb{E}_p[- \log p(z|y)]. \tag{9}
> $$
>
> For in-domain image compression, Eqs. (7), (8). and (9) hold, as $p(y|z)$, $p(z)$, and $p(z|y)$ are close to optimal probability representations $p(y^*|z^*)$, $p(z^*)$, and $p(z^*|y^*)$ due to the assumption of the optimal joint probability approximation of true marginal distribution.
> ***
>
>
> ***
> ***
> **W4:** In the proof of Proposition 2, the authors argue that the entropy bottleneck $p(z_{t})$ and the posterior $p(y_{t}|z_{t})$ are learned from the source dataset (L212, source image correlated) and may not work effectively. While this makes sense, it is not a theoretical analysis, in my point of view.
>
> **Response:** Proposition 2 aims to clarify the practical joint probability will deteriorate when the parameters of learned encoding distribution are fixed in latent refinement scenarios and domain shifts are incurred.
>
> It may be straightforward to induce the proof of this proposition based on commonly used conclusions and existing observations. Specifically, [A] uncovered $p(y_{t}|z_{t},\theta_{h_{e}}^{s})$ is inaccurate to encode $y_{t}$ due to source image-correlated $\theta_{h_{e}}^{s}$ and unreliable $z_{t}$. [B] observed the entropy bottleneck $p(z_{t})$ parameterized by $\varphi^{s}$ is quite poor at specializing and encoding $z_{t}$ for cross-domain images. **Therefore, the deterioration of independent distributions results in the deterioration of the joint distribution.**
>
> In conclusion, it may be sufficient to clarify Proposition 2 using commonly used conclusions and existing observations, rather than sophisticated theoretical analysis. A revised version of the proof of Proposition 2 can be found in the revised manuscript. We hope your confusion can be eliminated.
>
> [A]: Content adaptive optimization for neural image compression, CVPR,2019
>
> [B]: Learned Compression of Encoding Distributions, ICIP,2024

---

> ### Author Response · Authors · 2024-11-16
> **Thank you; Address your concerns [3/3]**
>
> ***
> ***
>
> **W5:** For Corollary 1, it is unclear whether $z^*$ and $y^*$ in equation 13 correspond to in-domain or cross-domain cases. In the cross-domain case with $z_{t}^*$ and $y_{t}^*$ as in Proposition 2, equation 13 is as follows: $\Delta H = H(y,z)-H(y_{t}^*)>H(z^*|y^*)$, which holds according to equations (11, 12) only if $H(y_{t}^*)=H(y)$ and $H(z^*|y^*)=H(z_{t}|y_{t})$. However, there is no guarantee for the condition.
>
> **Response:**
>
> First, since we focus on the cross-domain cases in Corollary 1, $z^*$ and $y^*$ in Eqs. (11) and (12) actually mean $z^*_{t}$ and $y^*_{t}$, respectively, which correspond to optimal probability representations (that always exactly match the true underlying distribution as clarified in Definition 1) with minimum rate cost. Moreover, $z$ and $y$ in Eqs. (11) and (12) actually mean $z_{t}$ and $y_{t}$, respectively.
>
> Second, based on the clarifications in the first point, the extra rate consumption of cross-domain marginalization approximation is $\Delta H = H(y_{t},z_{t})-H(y_{t}^*)$, rather than $\Delta H = H(y,z)-H(y_{t}^*)$ (formulated by you). Thus, there does not refer to the conditions $H(y_{t}^*)=H(y)$ and $H(z^*|y^*)=H(z_{t}|y_{t})$ you mentioned.
>
> Based on these clarifications, a clearer version of Corollary 1 is represented in the revised manuscript as follows. We hope your confusion can be eliminated.
> ***
> **Corollary 1.** The extra rate consumption of cross-domain marginalization approximation $\Delta H$ will be larger than that of in-domain marginalization approximation, *i.e.*,
> $\Delta H > \Delta H^*$, due to deteriorated joint probability.
>
> **Proof.**
> Due to distribution shifts and fixed suboptimal source-domain parameters $(\theta_{h_{e}}^{s}, \varphi^{s})$, Eqs. (11) and (12) in Prop. 1 can be further represented based on Prop. 2 as follows,
>
> $$\Bbb E_p[-\log p(y_{t}|z_{t},\theta_{h_{e}}^{s}) - (-\log p(y_{t}^*|z_{t}^*))] > 0, s.t. p(y_{t}|z_{t},\theta_{h_{e}}^{s})\neq p(y_{t}^*|z_{t}^*) \tag{11}$$
>
> $$\Bbb E_p[-\log p(z_{t}|\varphi^{s})- (-\log p(z_{t}^*)] > 0, s.t. p(z_{t}|\varphi^{s}) \neq p(z_{t}^*) \tag{12}$$
>
> where $p(y_{t}|z_{t},\theta_{h_{e}}^{s})$$\neq$$p(y_{t}^*|z_{t}^*)$ holds due to source image-correlated $\theta_{h_e}^s$ and unreliable $z_t$. $p(z_{t}|\varphi^{s})$ $\neq$ $p(z_{t}^*)$ holds due to poor specialization of entropy bottleneck $\varphi^s$. Let $p(y_{t}|z_{t},\theta_{h_{e}}^{s})$=$p(y_{t}|z_{t})$ and $p(z_{t}|\varphi^{s})$=$p(z_{t})$ for brevity. We have:
>
> $$\Delta H = \Bbb E_p[-\log p(y_{t}|z_{t}) - (-\log p(y_{t}^*|z_{t}^*))] + \Bbb E_p[-\log p(z_{t})- (-\log p(z_{t}^*))] + \Bbb E_p[- \log p(z_{t}^*|y_{t}^*)] > \Delta H^* \tag{13}$$
>
> Since optimal probabilities always match the true underlying distributions from Prop. 2, $p(z_{t}^*|y_{t}^*)$ is equivalent to $p(z^*|y^*)$ in Eq. 13 for optimal posterior distributions. Thus, the above equation holds and implies more rate consumption is potentially incurred in cross-domain scenarios.
> ***
> ***
> ***
>
> **W6:**  For proposed distribution regularization, it is unclear how $H(z^*|y^*)=-\Bbb E_{y^*,z^*\sim p(y^*,z^*)}[p(z^*|y^*)]$ leads to $-\log p(z_{t}^{m}|y_{t}^{m})$ in the optimization objective.
>
> **Response:** In a practical optimization scenario, the computation of $H(z^*|y^*)$ is **computationally infeasible since we cannot directly access the optimal posterior distribution** $p(z^*|y^*)$. To address this problem, $p(z_{t}^{m}|y_{t}^{m})$ serves as an **empirical approximation** of $p(z^*|y^*)$ that can be optimized directly using the objective. This approximation allows us to iteratively refine $z_{t}^{m}$ and $y_{t}^{m}$ to better align with the optimal posterior by minimizing the entropy of $p(z_{t}^{m}|y_{t}^{m})$.
>
> In conclusion, $H(z^*|y^*)$ is empirically approximated by $H(z_{t}^{m}|y^m_{t})=\Bbb E_{y_{t},z_{t}\sim p(y_{t},z_{t})}[-\log p(z_{t}^m|y_{t}^m)]$ due to the intractable access of optimal posterior distribution $p(z^*|y^*)$. In the revised manuscript, we clarify this point in Eq. (17).
> ***
> ***
>
> **W7:** The writing is a bit difficult to understand, especially in the discussion parts of both the methodology and experiment sections.
>
> **Response:** We have followed your suggestion to improve the readability of the discussion section. For example, we make the probability representation consistent between the first discussion part of the experiment section and the first discussion part of the methodology section. This can help you better understand Eq. (16) by the entropy curves of different probabilities in Figure 4. Please refer to the revised manuscript for more details.

---

> ### Comment · Reviewer_5FRR · 2024-11-24
> **Thank you for the authors' response.**
>
> Thank you for the authors' response.
> Your explanations and clarifications help my understanding of your paper.
> I list the remaining concerns and questions below.
>
> ### 2.
> It seems to me that the concern has not been fully resolved yet.
> I provide the cases in detail below in 3,5 about whether the expectations are properly considered.
>
> I understand that it's challenging to include all the notational details in the main paper due to page limitations and readability.
> It would be great if a detailed version with a full description could be included in the Appendix.
>
> ### 3.
> It seems to me that the (proof of) proposition 1 still contains many unclear parts.
> For instance, the exact notation for equation 7 will be (following the equation 5):
>
> $$
> E_{y,z \sim p(y|z)p(y)} [-\log p(y| z)] - E_{y^*, z^* \sim p(y^*| z^*)p(z^*)}[-\log p(y^*|z^*)] = 0
> $$
>
> The condition where the equality holds is $p(y|z) = p(y^*|z^*)$ and $p(z) = p(z^*)$.
> Thus, "s.t. $p(y|z) = p(y^*|z^*)$" is not complete condition.
> Also, the exact expression for the second term in equation 9 would be
>
> $$
> \Delta H^* = \mathbb{E}_{y^*, z^* \sim p(y^*, z^*)}[-\log p(z^*|y^*)]
> $$
>
> (Question) Given the definition of the "optimal probability represention," does Proposition 1 simply indicates that in the in-domain distribution, where $y=y^*$ and $z=z^*$ are achievable, the extra consumption can be defined as $\Delta H^* = H(y, z) - H(y^*) = H(y^*, z^*) - H(y^*) = H(y^*|z^*)$?
>
>
> ### 5.
> In equation 11 and 12, it seems that there are some misleading parts in terms of expectation.
> By the definition of marginal approximation (equations 4 and 5), the equation 11 seems to be (omitting $\theta_{h_e}^s$ for simplicity):
>
> $$
> E_{y_t, z_t \sim p(y_t|z_t)p(y_t)}[-\log p(y_t|z_t)] - E_{y_t^*, z_t^* \sim p(y_t^*|z_t^*)p(z_t^*)}[-\log p(y_t^*|z_t^*)]
> $$
>
> making it unclear whether it is greater than zero. If equation 11 represents:
>
> $$
> E_{y_t^*, z_t^* \sim p(y_t^*|z_t^*)p(y_t^*)}[-\log p(y_t=y_t^*|z_t=z_t^*) - (- \log p(y_t^*|z_t^*))] = D_{\text{KL}}(p(y_t^*|z_t^*)||p(y_t|z_t))
> $$
>
> then it is greater than zero, but this differs from the definition of marginal approximation (change in expectation).
> The same applies to equation 12.
>
> ### 6.
> In equation 13, as the authors explained, $y_t^*$ and $z_t^*$ represent the optimal latent encoding of the target domain, and the optimization seems to be performed over $p(y_t|z_t)$ and $p(z_t)$ (with learnable parameters $\theta_{h_e}$ and $\psi$).
> In this case, it seems unclear to me the motivation of optimizing $H(z_t^*|y_t^*)$ by approximating it into $H(z_t^m|y_t^m)$ in equation 17.
>
> ### 7.
> For me, Propositions 1 and 2 seem more like terminology definitions regarding marginal approximation and general interpretations of the cross-domain problem rather than theoretical analyses. If there are contributions I have missed, please provide clarification.
>
> (I use $E$ instead of $\mathbb{E}$ in some cases to resolve rendering issue.)

---

> ### Author Response · Authors · 2024-11-25
> **Thank you; Address your remaining concerns [1/3]**
>
> Thanks for your feedback. We took your remaining concerns and questions seriously.  Please see our point-to-point responses as follows.
>
> **W2:**  It would be great if a detailed version with a full description could be included in the Appendix.
>
> **Response:** Thanks for your suggestions. We will follow your advice to provide detailed version in the Appendix (refers to revised paper).
> ***
> **W3-1:** the (proof of) proposition 1 still contains many unclear parts, s.t., $p(y|z)=p(y^*|z^*)$ is not complete condition.
>
> **Response:**  We agree with your opinion that making the condition more complete is necessary. Thus,
> we have followed your advice to revise the condition as $p(y|z)=p(y^*|z^*)$ and $p(z)=p(z^*)$.
> ***
> **W3-2** the exact expression for the second term in equation 9 would be $\Delta H^* = \Bbb E_{y^*,z^* \sim p(y^*,z^*)}[-\log p(y^*|z^*)]$.
>
> **Response:** We agree with your statement. We have followed your suggestion to revise the manuscript.
> ***
> **W3-3:** (Question) Given the definition of the "optimal probability represention," Does Proposition 1 simply indicates that in the in-domain distribution, where $y=y^*$ and $z=z^*$ are achievable, the extra consumption can be defined as $\Delta H^* = H(y,z) - H(y^*) = H(y^*,z^*) - H(y^*) =  H(y^*|z^*)$?
>
> **Response:** We kindly appreciate your derivation, but the correct one would be $\Delta H^* =  H(z^*|y^*)= H(z|y)$, which also aligns your original comment, *i.e.,* "... the proposition simply interprets $H(y,z) - H(y^*)=H(z|y)+H(y)-H(y^*)=H(z|y)$ when $H(y)=H(y^*)$...".  As we discussed there, while the decomposition of the joint distribution you suggested is feasible, leading to $H(y,z)=H(z|y)+H(y)$ related to $p(z|y)$ and $p(y)$, such decomposition may not directly align the R-D objective in Eq. 3, *i.e.,* $p(y|z)$ and $p(z)$ that are more meaningful in the neural compression community due to their direct correlations with rate cost. More importantly, $p(y|z)$ and $p(z)$ help us analyze the rate cost change of learned encoding distributions $p(y|z,\theta_{h_{e}})$ and $p(z|\varphi)$ when $\theta_{h_{e}}$ and $\varphi$ are fixed in latent refinement scenarios and domain shifts are incurred on cross-domain scenarios.
>
> To summarize, the motivation of our derivation in Proposition 1 focuses on the decomposition process of $p(y,z)$ and $p(y^{*})$, since the induced $p(y|z)$ and $p(z)$ can help us analyze the rate cost change of learned encoding distributions $p(y|z,\theta_{h_{e}})$ and $p(z|\varphi)$ when $\theta_{h_{e}}$ and $\varphi$ are fixed in latent refinement scenarios and domain shifts are incurred on cross-domain scenarios.
> ***

---

> ### Author Response · Authors · 2024-11-25
> **Thank you; Address your remaining concerns [2/3]**
>
> **W5:** ..., the equation 11 seems to be $\Bbb E_{y,z\sim p(y_{t}|z_{t})p(z_{t})}[-\log p(y_{t}|z_{t})] - \Bbb E_{y^*,z^*\sim p(y_{t}^*|z_{t}^*)p(z_{t}^*)}[-\log p(y_{t}^*|z_{t}^*)] $, making it unclear whether it is greater than zero.
>
> **Response:** Since $p(y_{t}^*|z_{t}^*)$ as the **optimal distribution representation**  (claimed in Proposition 2) can achieve the theoretical minimum coding length that equals the entropy of the true data distribution, $\Bbb E_{y^*,z^*\sim p(y_{t}^*|z_{t}^*)p(z_{t}^*)}[-\log p(y_{t}^*|z_{t}^*)]$ corresponds to the lower bound of the entropy consumption of coding distribution according to Shannon’s entropy theorem[A].
>
> Therefore, it is feasible to make $\Bbb E_{y,z\sim p(y_{t}|z_{t})p(z_{t})}[-\log p(y_{t}|z_{t})] - \Bbb E_{y^*,z^*\sim p(y_{t}^*|z_{t}^*)p(z_{t}^*)}[-\log p(y_{t}^*|z_{t}^*)] > 0$ hold by leveraging Shannon’s entropy theorem (as described in Definition 1) and the suboptimal condition $\quad p(y_{t}|z_{t},\theta_{h_{e}}^s)\neq p(y_{t}^*|z_{t}^*)$ of coding distribution on cross-domain scenarios (as claimed in Proposition 2). Specifically,
> ***
> According to Shannon's entropy theorem, the optimal probability distribution $p(y_t^*|z_t^*)$ achieves the theoretical minimum coding length, which equals the entropy of the true data distribution. Any non-optimal coding distribution (e.g., $p(y_t|z_t,\theta_{h_{e}}^{s}))$ must have a higher expected code length than the theoretical minimum. This can be reflected by the entropy consumption of coding distribution to make the following inequality strictly hold:
>
> \begin{eqnarray}
>     \Bbb E_{y,z\sim p(y_t|z_t)p(z_t)}[-\log p(y_t|z_t,\theta_{h_{e}}^{s})] \geq \Bbb E_{y^*,z^*\sim p(y_t^*|z_t^*)p(z_t^*)}[-\log p(y_t^*|z_t^*)],
> \end{eqnarray}
>
> where the equality holds only if $p(y_t^*|z_t^*) = p(y_t|z_t,\theta_{h_{e}}^{s})$ and $p(z_t|\varphi^{s}) = p(z_{t}^*)$.
>
> Since we have assumed that $p(y_t|z_t,\theta_{h_{e}}^{s}) ≠ p(y_t^*|z_t^*)$ holds due to source image-correlated $\theta_{h_{e}}^{s}$ and unreliable $p(z_t|\varphi^{s}) ≠ p(z_{t}^*)$ on cross-domain scenarios. Thus,
>
> \begin{eqnarray}
>     \Bbb E_{y,z\sim p(y_t|z_t)p(z_t)}[-\log p(y_t|z_t,\theta_{h_{e}}^{s})] > \Bbb E_{y^*,z^*\sim p(y_t^*|z_t^*)p(z_t^*)}[-\log p(y_t^*|z_t^*)]
> \end{eqnarray}
>
> holds. Thus, the following inequality can be naturally derived,
>
> \begin{eqnarray}
>     \Bbb E_{y,z\sim p(y_t|z_t)p(z_t)}[-\log p(y_t|z_t,\theta_{h_{e}}^{s})] - \Bbb E_{y^*,z^*\sim p(y_t^*|z_t^*)p(z_t^*)}[-\log p(y_t^*|z_t^*)] > 0.
> \end{eqnarray}
>
> To summarize, it is clear equation 11 holds. The same applies to equation 12. Under the guidance of optimal distribution representation corresponding to minimum entropy consumption ( minimum coding length),  we hope this explanation can eliminate your confusion.
> ***
>
> **W6-1:**  ..., and the optimization seems to be performed over $p(y_{t}|z_{t})$ and $p(z_{t})$ (with learnable parameters $\theta_{h_{e}}$ and $\varphi$).
>
> **Response:** **Regarding the statement "the optimization seems to be performed over $p(y_{t}|z_{t})$ and $p(z_{t})$ (with learnable parameters $\theta_{h_{e}}$ and $\varphi$", it should be noted that learnable parameters $\theta_{h_{e}}$ and $\varphi$ are fixed without any updates during the optimization process in Eq. (17),** since our method is constructed by latent refinement framework (refers to line 250-253) and Eq. (15) (only latent representations are updated).
>
> [A] Bromiley, P. A., N. A. Thacker, and E. Bouhova-Thacker. "Shannon entropy, Renyi entropy, and information." Statistics and Inf. Series (2004-004) 9.2004 (2004): 2-8.

---

> ### Author Response · Authors · 2024-11-25
> **Thank you; Address your remaining concerns [3/3]**
>
> **W6-2:** In this case, it seems unclear to me the motivation of optimizing $p(z_{t}^*|y_{t}^*)$ by approximating it into $p(z_{t}^m|y_{t}^m)$ in equation 17.
>
>
> **Response:**
> $$
> L_{DR}= \Bbb  E_{x_{t}\sim p_{x_{t}},y_{t},z_{t}\sim p_{y_{t},z_{t}}} [-\log p(y_{t}^{m}|z_{t}^{m},\theta_{h_{e}}^{s}) -\log p(z_{t}^{m}|\varphi^{s}) + \beta(- \log p(z_{t}^{m}|y_{t}^{m}))  + \lambda (-\log p(x_{t}|y_{t}^{m}))] \tag{17}
> $$
>
> The motivation of optimizing $p(z_{t}^*|y_{t}^*)$ by approximating it into $p(z_{t}^{m}|y_{t}^{m})$ in equation (17) includes two perspectives:
>
>
> - If the balance coefficient $\beta$ is  1, minimizing the entropy of probability estimates  ($i.e.,$ the first two terms in Eq. (17) is equivalent to minimizing the distribution gap between estimated probability $p(y|z)$ or $p(z)$ and (unknown) optimal probability $p(y^*|z^*)$ or $p(z^*)$ [B]. This coincides with the objectives of the first two terms in Eq. (13) of Cor. 1, *i.e.,*
> $$\Delta H = \Bbb E_p[-\log p(y_{t}|z_{t}) - (-\log p(y_{t}^*|z_{t}^*))] + \Bbb E_p[-\log p(z_{t})- (-\log p(z_{t}^*))] + \Bbb E_p[- \log p(z_{t}^*|y_{t}^*)] > \Delta H^* \tag{13}$$
> **Meanwhile, since the optimal posterior distribution** $p(z_{t}^*|y_{t}^*)$ **cannot be accessed**,  **using its empirical approximation** $p(z_{t}^{m}|y_{t}^{m})$ **as an alternative can make Eq. (17) be equivalent to**
> $$
>     L_{DR} \propto \Delta H + \lambda (-\log p(x_{t}|y_{t}^{m})). \tag{18}
> $$
> By recalling Eq. (13) in Cor. 1 and Eq. (14) ($\Delta H^M > \Delta H^0 >\Delta H^* $), the proposed distribution regularization and **corresponding empirical approximation** can encourage the deteriorated joint probability approximation to approach the initial and even optimal ones, leading to lower extra rate consumption of marginalization approximation. In other words, $\Delta H^*$ is the **lower bound** of the first term of Eq. (18), *i.e.*, the better joint probability approximation, the closer to the lower bound. $L_{DR}$ can potentially remedy the additional rate of consumption of vanilla HLR.
>
> - **For the optimization process in the vanilla HLR,  there is no explicit constraint to ensure that** $z_{t}^{m}$ **can well match its posterior distribution under the condition of** $y_{t}^{m}$, **leading to deteriorated joint probability.** **As an empirical approximation of the ideal posterior distribution** $p(y_{t}^*|z_{t}^*)$, **the proposed distribution regularization** $p(z_{t}^{m}|y_{t}^{m})$ **can eliminate such an issue.**
> ***
> **W7:** For me, Propositions 1 and 2 seem more like terminology definitions regarding marginal approximation and general interpretations of the cross-domain problem rather than theoretical analyses. If there are contributions I have missed, please provide clarification.
>
> **Response:**
> - **Our core contribution lies in the proposed distribution regularization**, which encourages the deteriorated joint probability approximation to approach both the initial and optimal distributions, leading to lower extra rate consumption in marginalization approximation. **Propositions 1, 2, Corollary 1, and related analyses help readers understand the origins of our proposed distribution regularization and explain why it reduces extra rate consumption. Thus, these propositions, corollary, and related analyses are necessary and meaningful.**
>
>
> - To the best of our knowledge, we are also the first to uncover the rate degradation reason of joint probability approximation of the marginal distribution from in- to cross-domain, exhibiting the contribution.
>
>
> **To summarize, we agree that Propositions 1 and 2 seem more like terminology definitions regarding marginal approximation and general interpretations of the cross-domain problem. Instead of providing strictly theoretical theorem, our paper has made the first attempt to uncover the rate degradation reason of joint probability approximation of the marginal distribution from in- to cross-domain by detailed analysis and  perspectives in Propositions 1, 2, and Corollary 1.**  More importantly, we leverage these analyses to help readers understand the origins of our proposed distribution regularization and explain why it reduces extra rate consumption, exhibiting the core contribution of this paper.
>
> [B]: David JC MacKay. Information theory, inference and learning algorithms. Cambridge university press, 2003.

---

> > ### Author Response · Authors · 2024-12-01
> > **Thank you**
> >
> > Dear Reviewer 5FRR,
> >
> > As the discussion period is ending soon, we would like to send a kind reminder about our latest responses and the revised manuscript. We have addressed each of your concerns in detail and incorporated your suggestions into our revisions. If there are still remaining concerns, we will do our best to provide clarifications as soon as possible. Otherwise, we look forward to your positive feedback.
> >
> > Once again, we appreciate your time and consideration.

---

> ### Comment · Reviewer_5FRR · 2024-12-02
> **Thank you for the authors' detailed responses.**
>
> Thank you for the authors' detailed responses.
> I have reviewed your response and the revised version of the paper.
> The responses have addressed most of my concerns and have helped my understanding of the method.
> I now agree with the analysis presented in Propositions 1 and 2, as well as Corollary 1, regarding the shortcomings of vanilla HLR.
> Although the derivation of the regularization term is not entirely clear, it seems reasonable that the term itself can be helpful to HLR.
> Since your responses have addressed most of my concerns, I have raised the score.

---

> ### Author Response · Authors · 2024-12-02
>
> Dear Reviewer 5FRR,
>
> Thanks for raising your score and acknowledging our contributions.
> We appreciate your valuable suggestions that have intensely improved our paper.
>
> Thanks for your time again.
>
> Best wishes,
>
> The authors

---

### Author Response · Authors · 2024-12-02
**Thanks for your time**

Dear Reviewers,

As the discussion period will be ended today, we would like to send a kind reminder about our latest responses for Reviewer 5FRR (regarding theoretical discussions) and Reviewer ShHq (regarding the concern of “insufficient comparisons”) .

We have addressed your concerns in detail via responses or incorporated your suggestions into our revisions.
We are looking forward to your feedback.

Best wishes,

The authors

---

### Meta-Review · Area_Chair_BQu4 · 2024-12-19

**Metareview:**

This paper addresses test-time adaptation for learned image compression (TTA-IC) in the face of cross-domain inputs. It builds upon a latent-refinement approach originally designed for in-domain scenarios and identifies a key issue: a fundamental mismatch between the refined Gaussian conditionals and hyperprior distributions when adapting to out-of-distribution images. This mismatch leads to suboptimal joint probability approximations and increased bit rates. To remedy this, the authors introduce a Bayesian approximation-based distribution regularization, which better aligns latent distributions with their hyperpriors, resulting in improved rate-distortion (R-D) performance on cross-domain inputs.

Initial reviewer feedback was mixed. Major concerns included:

1. **Clarity and Accessibility:** The motivations and theoretical derivations were challenging to follow, especially for readers without substantial background in learned compression.
2. **Comparisons to General TTA Methods:** The paper lacked explicit connections and comparisons to broader test-time adaptation techniques.
3. **Computational Complexity:** Reviewers requested more details on computational overhead and complexity.
4. **Dataset Diversity:** The evaluation lacked experiments on a broader range of datasets.

During the rebuttal phase, the authors provided detailed clarifications, introduced additional experiments on more diverse datasets, and offered complexity analyses. These responses addressed most concerns, and by the end of the discussion, all reviewers agreed on a borderline acceptance.

The AC read paper, reviews, and discussions, and concurs with the reviewers. The additional clarifications and experiment results presented in the rebuttal adequately addresses the reviewers’ concerns. However, the AC also agrees with the reviewers that the paper can be benefited from a significant revision by improving its clarity and self-containedness; many of the mathematical derivations are either trivial or from the prior works, and can be greatly simplified  (e.g., Section 3.1.). Introducing more descriptions on background (e.g., what information the side-information is carrying and be used in decoding) can make the paper more easily accessable to non-experts of the domain. Overall, the AC believes that the paper provides a meaningful theoretical insight and demonstrates tangible performance improvements, hence recommends acceptance. The authors are encouraged to incorporate the clarifications and new results presented in the rebuttal into the camera-ready version for better clarity and accessibility.

**Additional Comments On Reviewer Discussion:**

The paper initially received several critical comments by the reviewers such as clarification on presentation and derivations (5FRR, ShHq, 7uht) and insufficient comparisons (ShHq, GYDX). These are adequately addressed by the authors in the rebuttal, leading the reviewers to recommend acceptance.

---

### Decision · Program_Chairs · 2025-01-22

Accept (Poster)